# NEUROTREE: Hierarchical Functional Brain Pathway Decoding for Mental Health Disorders

**Jun-En Ding**[1] **Dongsheng Luo**[2] **Chenwei Wu**[3] **Feng Liu**[1]

## Abstract

Mental disorders are among the most widespread diseases globally. Analyzing functional brain networks through functional magnetic resonance imaging (fMRI) is crucial for understanding mental disorder behaviors. Although existing fMRI-based graph neural networks (GNNs) have demonstrated significant potential in brain network feature extraction, they often fail to characterize complex relationships between brain regions and demographic information in mental disorders. To overcome these limitations, we propose a learnable NeuroTree framework that integrates a $k$-hop AGE-GCN with neural ordinary differential equations (ODEs) and contrastive masked functional connectivity (CMFC) to enhance similarities and dissimilarities of brain region distance. Furthermore, NeuroTree effectively decodes fMRI network features into tree structures, which improves the capture of high-order brain regional pathway features and enables the identification of hierarchical neural behavioral patterns essential for understanding disease-related brain subnetworks. Our empirical evaluations demonstrate that NeuroTree achieves state-of-the-art performance across two distinct mental disorder datasets. It provides valuable insights into age-related deterioration patterns, elucidating their underlying neural mechanisms. The code and datasets are available at https://github.com/Ding1119/NeuroTree.

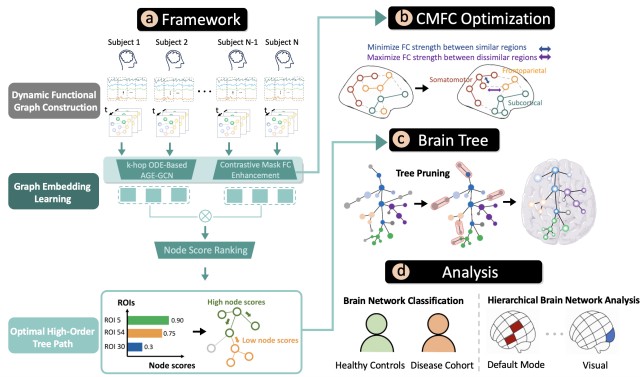

*Figure 1.* Overview of NEUROTREE framework.

## 1. Introduction

In recent years, neuropsychiatric disorders and addiction have emerged as substantial public health challenges, with their impact on brain function becoming a critical research focus in neuroscience (Sahakian et al., 2015; Hollander et al., 2020; Borumandnia et al., 2022). Substance use disorder (SUD) is widely studied in the field of mental disorders, enabling the investigation of structural brain alterations among SUD users across various factors (e.g., age or gender) (Niklason et al., 2022; Ding et al., 2024b). The development of brain imaging modalities, such functional magnetic resonance imaging (fMRI), which measures blood oxygen level-dependent (BOLD) signals, and electroencephalogram (EEG), which measures the electrical potentials of the brain (Jiao et al., 2024; Yang et al., 2024; Yang & Liu, 2024) has provided a means to analyze brain network activities in mental disorders (Zhang et al., 2016; Li et al., 2022a; Niklason et al., 2022; Pan et al., 2025). Moreover, research on the clustering of connectivity in brain regions using fMRI has demonstrated that the brain does not operate statically but instead exhibits a dynamic, multi-level hierarchical organization (Meunier et al., 2009; Betzel et al., 2023). In particular, hierarchical network analysis has been shown to effectively distinguish these abnormal patterns and identify brain regions associated with specific pathological mechanisms in psychiatric disorders (Boisvert et al., 2024; Segal et al., 2023; Wen et al., 2023b).

Currently, traditional brain network analysis has success-

---

[1]Department of Systems Engineering, Stevens Institute of Technology, New Jersey, USA [2]Department of Computing and Information Sciences, Florida International University, USA [3]Department of Electrical Engineering and Computer Science, University of Michigan, USA. Correspondence to: Feng Liu <fliu22@stevens.edu>.

*Proceedings of the $42^{nd}$ International Conference on Machine Learning*, Vancouver, Canada. PMLR 267, 2025. Copyright 2025 by the author(s).

fully represented regions of interest (ROIs) as nodes with functional connectivity (FC) as edges (Kulkarni et al., 2023; Xu et al., 2024; 2016). Specifically, research on graph neural networks (GNNs) has shown promising results, demonstrating strong predictive capabilities for neuropsychiatric disorders by learning patterns across brain networks (Tong et al., 2023; Kazi et al., 2019).

Differential equations can capture the dynamic processes underlying changes in brain activity over time. CortexODE further advances these approaches by combining ordinary differential equations (ODEs) with deep learning methods to model MRI images as point trajectories, thereby enabling the reconstruction of complex cortical surfaces (Ma et al., 2022). Moreover, STE-ODE (Tang et al., 2024) combines the advantages of ODEs with continuous-time fMRI and structural MRI to model dynamic brain networks while leveraging GNN architectures for embedding learning. Currently, the existing methods have several limitations. First, although GNNs demonstrate excellent node feature learning capabilities in graph structures, they are generally constrained by limited interpretability—particularly regarding the explicability of auxiliary factors underlying mental disorders. Second, graph-based networks for human brain FC are restricted to feature learning from dynamic graph sequences, overlooking the brain's hierarchical clustering of network activity. Consequently, there is a lack of effective methodologies for predicting functional brain alterations across different age stages in psychiatric disorders.

To address these limitations, we leverage the properties of ODEs for fMRI modeling similar to STE-ODE and extend continuous-time dynamics methods. In this study, we propose NEUROTREE, a framework designed to characterize brain network patterns through hierarchical analysis and represent the fMRI network as tree-structured pathways. This approach decodes brain network features and facilitates visualization of disorder propagation through brain pathways. Additionally, we develop a $k$-hop ODE-based AGE-GCN to capture complex interactions between adjacent and distant brain regions and to measure the convergence of various mental disorder subtypes. We incorporate demographic features like age into model learning and message passing to enhance interpretability. Moreover, we implement contrastive masking optimization to identify key FC patterns. Our main contributions are as follows:

- We integrate demographic information into Neural ODEs for modeling both static and dynamic brain networks via $k$-hop graph convolution.

- We investigate two types of brain disorder datasets to decode fMRI signals, constructing disease-specific brain trees while employing trainable framework to identify hierarchical functional subnetwork regions.

- Our research not only achieves state-of-the-art graph

classification performance but also effectively interprets how addiction and schizophrenia disorders cause changes in FC related to brain age.

## 2. Related Work

**Neural ODEs for Dynamic BOLD fMRI Signal Modeling.** The BOLD fMRI signal is an indirect and delayed reflection of neural activity. Traditional approaches like DCM use ordinary differential equations (ODEs) to infer connectivity, but they are computationally intensive and rely on prior assumptions (Cao et al., 2019). Recent work has demonstrated the effectiveness of ODEs in modeling continuous-time neural dynamics (Han et al., 2024; Tang et al., 2024). The evolution of brain states can be described by the equation $\frac{dX(t)}{dt} = f(X(t), t)$, where $X(t)$ represents the brain state at time $t$, and $f(\cdot)$ is a neural network that learns the rules governing state transitions. This framework provides a natural way to model the continuous nature of brain dynamics while maintaining biological plausibility (Havlicek et al., 2015).

**Path-Aware Learning in GNNs.** GCKN computes node distances using kernel functions along vertex-connected paths (Chen et al., 2020). Additionally, PathNNs enhance the expressive power of GNNs by aggregating information from multiple paths between nodes to improve performance across multiple graph classification and regression tasks and capturing complex graph structures (Michel et al., 2023). The HEmoN model (Huang et al., 2025) has provided preliminary insights into human emotions through tree path analysis; however, it lacks aggregation of higher-order brain pathways in complex brain region interactions, as well as consideration of different demographic factors between individuals. In this study, we propose a weighted higher-order brain tree pathway framework that enhances pathway representation and effectively explains the predictions of changes in FC networks associated with mental disorders across different age groups.

## 3. The Proposed Architecture

The proposed NEUROTREE framework, illustrated in Fig. 1, constructs dynamic FC graphs from subjects' time-series data. It encodes dynamic graph fMRI features using a $k$-hop ODE-Based AGE-GCN incorporate demographics for graph embedding learning for downstream tasks (e.g., graph classification, brain age prediction). Subsequently, a CMFC optimization is performed to adjust the FC strength between similar and dissimilar brain regions distance. Then, we build the hierarchical brain tree by reranking predicted brain regions of interest (ROIs) scores and high-order tree path weights, followed by tree pruning to create anatomically meaningful hierarchical brain network structures. The resulting brain tree structures are then employed to interpret

disease cohorts, where functional networks primarily affecting mental disorders manifest at different tree levels.

## 4. Preliminaries

**Problem Formulation and Notations.** In this section, we formulate the classification of mental disorders using both static and sequential segmented instance brain networks constructed from fMRI as a supervised graph classification task for each patient. Let temporal brain activity measurements from $N$ patients, $X_i \in \mathbb{R}^{v \times T}$, represent the complete BOLD fMRI signals with $v$ regions of interest (ROIs) and a total length of $T$ time series. We denote $X_i(t) \in \mathbb{R}^{v \times T'}$ is segmented time points with dimension $T'$ for the $i$th subject. For each subject, we construct a static or dynamic FC graph $G(\mathcal{V}, \mathcal{E})$, where $\mathcal{V}$ represents the set of vertices (i.e., brain regions) and $\mathcal{E}$ denotes the set of edges (i.e., functional connections). The FC between regions is typically computed using Pearson correlation methods. Based on these temporal correlation patterns, we derive both a static adjacency matrix $A^s \in \mathbb{R}^{v \times v}$ from the complete time sequence and time-varying dynamic FC matrices $A^d(t) \in \mathbb{R}^{v \times v}$. The static matrices capture averaged temporal dependencies, while the dynamic matrices preserve the temporal evolution of neural interactions (Zhang et al., 2016; Li et al., 2022a).

### 4.1. Graph Convolutional Networks

Traditional GCNs operate through neighborhood aggregation, whereby each node's representation is updated by aggregating features from its adjacent nodes. The classical GCN layer is formulated as

$$H^{(l)} = \sigma(D^{-\frac{1}{2}} A D^{-\frac{1}{2}} H^{(l-1)} W^{(l-1)}). \quad (1)$$

where $A$ is the adjacency matrix, $D$ is the degree matrix, $H^{(l)}$ represents the node features at layer $l$, and $W^{(l)}$ contains learnable parameters. However, this formulation is limited to first-order neighborhoods and may not effectively capture the complex interactions among brain region (Veličković et al., 2017).

### 4.2. $K$-hop ODE-Based Graph Embedding

The ODE effectively captures the long-term dependencies of dynamic changes in effective connectivity between different ROIs (Sanchez-Romero et al., 2019). Inspired by existing studies (Cao et al., 2019; Wen et al., 2024; Tang et al., 2024) that utilize ODEs to model BOLD signals or biomarkers in Alzheimer's disease (AD) research, we extend the ODE modeling framework by incorporating a scalar parameter, $\theta \in \mathbb{R}^+$ (i.e., age) , representing the chronological age of subjects. This enhancement aims to improve the prediction and interpretability of mental disorders in fMRI analysis.

$$\dot{X} = f(X(t), u(t), \theta) \quad (2)$$

where $u(t)$ denotes the external experimental stimuli. We consider the influence of an external parameter, $\theta$, on FC across the entire temporal time series $X(t)$, which is represented by adjacency matrix $A^d$. The following ODE models this relationship:

$$\frac{dX(t)}{dt} = \eta A^d(t) X(t) + \rho \cdot \theta X(t) + C u(t), \quad (3)$$

where the matrix $C \in \mathbb{R}^{v \times v}$ encodes the reception of external stimuli $u(t) \in \mathbb{R}^v$ and parameters $\eta, \rho \in \mathbb{R}$. To simplify the analysis, we assume that $C u(t) = 0$, indicating the absence of additional external stimuli due to resting-state fMRI task. By discretizing Eq. 3, we can derive the effective connectivity matrix $A^d(t)$. Its discrete representation is given by:

$$A^d(t) = \frac{1}{\eta} \left( \frac{dX(t)}{dt} \frac{1}{X(t)} - \rho \cdot \theta \right) \quad (4)$$

$$\overset{\Delta t = 1}{\approx} \varphi \left( \frac{X(t+1) - X(t)}{X(t)} - \rho \cdot \theta \right) \quad (5)$$

where $\varphi = \frac{1}{\eta}$ and $\rho$ are scale factors ranging from 0 to 1. To address the limitations of first-order feature convolution filters in traditional GCNs, we propose an extended spatial feature filter that operates over a $k$-hop connectivity operator. This approach enables the spatial $k$-order brain network representation to be learned via a generalized graph convolution, as defined in definition 4.1.

**Definition 4.1** ($k$-hop graph convolution)**.** The spatial $k$-order brain network representation can be learned through a generalized graph convolution as follows:

$$H^{(l+1)}(t) = \sigma \left( \sum_{k=0}^{K-1} \Phi_k(t) H^{(l)}(t) W_k^{(l)} \right) \quad (6)$$

where $\sigma$ denotes the ReLU function, and $\Phi_k(t) = \hat{D}^{-\frac{1}{2}} \hat{A}_k(t) \hat{D}^{-\frac{1}{2}}$ represents the normalized $k$-hop connectivity adjacency operator. The adjacency operator can be defined as:

$$\hat{A}_k(t) = \Gamma \odot A^s \odot \underbrace{[\lambda A^d(t) + (1-\lambda)(A^d(t))^T]^k}_{\textbf{k-hop connectivity}}. \quad (7)$$

where $\Gamma$ is a learnable parameter, the $k$-hop connectivity operator retains directional information, as evidenced by the fact that $\hat{A}_k(t) \neq (\hat{A}_k(t))^T$ when $\lambda \neq \frac{1}{2}$. This result is derived using the binomial theorem, as detailed in Appendix A. Intuitively, Eq. 7 considers that $A^s$ provides connectivity information for the entire static fMRI sequence, while $A^d$ can regulate the bidirectional combination of dynamic fMRI in the brain through the parameter $\lambda$.

**Theorem 4.2.** *Suppose the $l^2$-norm of the $k$-hop connectivity adjacency operator $\|\hat{A}_k(t)\|_2$ can be derived from Eq. (7). Then, as the $k$-hop approaches infinity ($k \to \infty$), the $\|\hat{A}_k(t)\|_2$ is bounded by the following inequality:*

$$\lim_{k \to \infty} \|\hat{A}_k(t)\|_2 \leq \|\Gamma\|_2 \cdot \|A^s\|_2 \cdot \max(\lambda, 1 - \lambda)^k \quad (8)$$

The detailed proof of Eq. 8 is provided in Appendix B. To ensure numerical stability when computing powers of $\hat{A}_k(t)$, we establish the bounded condition: $\|\Phi_k(t)\|_2 \leq 1$, which holds for all values of $k$ and $t$. In proposition 4.3, we demonstrate that $\Phi_k(t)$ exhibits smooth temporal evolution of the graph structure while maintaining a well-defined upper bound.

**Proposition 4.3** (Convergence and Uniqueness). *Let $\Phi_k(t)$ be the normalized adjacency operator that satisfies the Lipschitz condition (Coddington & Levinson, 1955) $\|\Phi_k(t_1) - \Phi_k(t_2)\|_2 \leq L\|t_1 - t_2\|_2$ for some constant $L \geq 0$. Then, for a Lipschitz continuous activation function $\sigma$ and bounded weights $W_k^{(l)}$, the $k$-hop ODE-GCN admits a unique solution.*

The proof from Picard's existence theorem can be found in Appendix C.

**Theorem 4.4** (Discretization of Age-Aware Continuous–Time Graph Convolution (Tang et al., 2024)). *Consider a series of brain networks $\{\mathcal{G}(t; \theta)\}_{t=0}^{T}$ parameterized by an age variable $\theta$. Let $Z(t)$ denote the embedding (or feature representation) of the STE-ODE network at time $t$. The continuous temporal evolution of the brain network can be expressed through the following age-modulated ODE-based graph convolution representation:*

$$Z(t + \Delta t) = Z(t) + \int_{t}^{t+\Delta t} F(Z(\tau), \tau, \theta) \, d\tau, \quad (9)$$

*where $F(\cdot)$ is the graph embedding function that captures the spatiotemporal and age-related information within the network.*

Specifically, the temporal evolution function is defined as:

$$F(Z(t), t, \theta) = \sum_{l=1}^{L} H^{(l)}(t, \theta), \quad (10)$$

where $H^{(l)}(t, \theta)$ representing the age-modulated layer-wise transformation at time $t$. Using a first-order approximation, the temporal dynamics described in Eq. 6 and theorem 4.4 can be discretized into the following form:

$$Z(t + \Delta t) \approx Z(t) + \Delta t \cdot F_\theta(Z(t))$$
$$= Z(t) + \Delta t \sum_{l=1}^{L} \sigma\Big(\sum_{k=0}^{K-1} \Phi_k(t) X_\theta^{(l)}(t) W_k^{(l)}\Big), \quad (11)$$

Here, $X_\theta^{(l)}(t) = X^{(l)}(t) \odot (\beta \cdot \theta)$ denotes the age-modulated feature matrix with learnable parameter $\beta \in (0, 1)$ for age modulation strength at layer $l$, and $W_k^{(l)}$ represents the learnable weights associated with the $k$-th hop filter weight at layer $l$.

For $t \geq 1$, the temporal age-modulated graph embedding update is expressed as:

$$Z(t + 1) = Z(t) + \text{AGE-GCN}_k(\mathcal{G}^f(t+1), \theta)$$
$$= Z(t) + \sigma\Big(\sum_{k=0}^{K-1} \Phi_k(t+1) X_\theta^{(L)}(t+1) W_k^{(L)}\Big)$$
$$(12)$$

where $\text{AGE-GCN}_k(\mathcal{G}^f(t+1), \theta)$ denotes the age-modulated $k$-hop graph convolution applied to the graph $\mathcal{G}^f(t+1)$ at the final layer $L$.

### 4.3. Optimization of Objective Function

To enhance the distinction between brain regions, we developed a node-wise contrastive masked functional connectivity (CMFC) loss function that simultaneously minimizes similarities and maximizes dissimilarities. This loss effectively amplifies the distances between intra-FCstrengths. For each node $i$, we calculate the FC strength $C_i(t) = \frac{1}{|\mathcal{V}|} \sum_{j=1}^{|\mathcal{V}|} |A_j^d(t)|$ at timestamp $t$, where $|\mathcal{V}|$ represents the total number of nodes in the network. We then map this FC strength to the latent space using a multilayer perceptron $h_i(t) = \text{MLP}(C_i(t))$. Subsequently, we generate positive and negative masks for connectivity pairs according to $M_{ij}^{\pm}(t) = \mathbb{I}\big((C_i^d(t) \gtrless \mu^C(t)) \wedge (C_j^d(t) \gtrless \mu^C(t))\big)$, where $\mu^C(t)$ is the mean value of FC and $\mathbb{I}(\cdot)$ is the indicator function. The positive and negative sample sets are then defined as $\mathcal{A}^{\pm}(t) = \{(i, j) \mid M_{ij}^{\pm}(t) = 1\}$. Finally, the CMFC loss is expressed as:

$$\mathcal{L}_{\text{pos}} = -\frac{1}{|\mathcal{A}^+|} \sum_{(i,j) \in \mathcal{A}^+} \log\Big(\frac{\exp(S_{ij}(t))}{\sum_{k \in \mathcal{V}} \exp(S_{ik}(t)) + \epsilon}\Big),$$
$$\mathcal{L}_{\text{neg}} = -\frac{1}{|\mathcal{A}^-|} \sum_{(i,j) \in \mathcal{A}^-} \log\Big(1 - \frac{\exp(S_{ij}(t))}{\sum_{k \in \mathcal{V}} \exp(S_{ik}(t)) + \epsilon}\Big).$$
$$(13)$$

where $S_{ij}(t) = \frac{\langle h_i(t), h_j(t) \rangle}{\|h_i(t)\| \|h_j(t)\|}$ computes the cosine similarity of latent vectors of FC for nodes $i$ and $j$ at time $t$, and $\epsilon = 10^{-6}$ is added to prevent division by zero. To optimize the model, we minimize the combined CMFC loss $\mathcal{L}_{CMFC} = \mathcal{L}_{\text{pos}} + \mathcal{L}_{\text{neg}}$ and combine it with the binary loss $\mathcal{L}_b$ for the overall loss function $\mathcal{L} = \mathcal{L}_{CMFC} + \mathcal{L}_b$.

**Node Score Predictor**

Different mental disorders exhibit distinct intrinsic FC patterns between various brain regions (Gallo et al., 2023; Fu et al., 2021). By analyzing these connectivity differences

in specific brain areas, we can predict regional scores to identify similar abnormal functional networks (Li et al., 2020). We leverage FC strength latent vectors $h_i$ combined with age-aware $k$-hop graph embeddings parameterized by $\Theta = \{\Lambda, \theta, W_k^{(l)}\}$ to generate predictive scores for individual brain regions. The score for the $i$th node is expressed as follows:

$$\mathcal{S}_i = h_i \cdot \zeta\left(\frac{1}{|\mathcal{V}|}\sum_{j \in \mathcal{V}} Z_j(\Theta)^\top Z_i(\Theta)\right), \quad i \in \{1, 2, \ldots, |\mathcal{V}|\} \tag{14}$$

where $\zeta(\cdot)$ denotes a non-linear transformation function and $Z(\Theta) \in \mathbb{R}^{|\mathcal{V}| \times d}$ represents the $d$-dimensional node embeddings learned by AGE-GCN with the parameter set $\Theta$.

**Definition 4.5** (MST algorithm). Given a weighted undirected graph $G^u = (V, E, w^u)$, where $w^u(e_{ij})$ assigns a weight to the edge $e_{ij} \in E$ between vertices $i$ and $j$, we define the pruned weighted graph $\mathcal{T}(G^u) = (\mathcal{V}, \mathcal{E}_\mathcal{T}, w^u)$, where $\mathcal{E}_\mathcal{T} \subseteq E$, $|\mathcal{E}_\mathcal{T}| \ll |E|$ represents the set of edges retained in the minimum spanning tree (MST) after pruning. Our goal is to find the minimum spanning tree while satisfying:

$$\min_{T \in \mathcal{T}(G^u)} \sum_{e_{ij} \in \mathcal{E}_T} w^u(e_{ij}). \tag{15}$$

where $T$ is a spanning tree with edges $\mathcal{E}_T \subseteq \mathcal{E}$ and diameter upper bound $\text{diam}(T) \leq \frac{(|\mathcal{V}|-1)(|\mathcal{V}|-2)}{2}$ (Spira & Pan, 1975). In practice, we can apply Kruskal's algorithm (Kruskal, 1956) to find the minimum spanning tree.

## 5. Hierarchical Brain Tree Construction

To characterize the hierarchical levels of FC in brain networks associated with mental illness, we construct a tree structure from a weighted undirected FC graph $G^u$ based on predicted node scores and weighted edges (i.e., FC strength), which contributes to identifying the most significant functional pathways. Note that Eq. 7 represents a model trained on directional graphs to investigate complex asymmetric connection patterns for graph embedding. The constructed tree of edge weights $w^u(e_{ij})$ is derived from the model's learnable FC strengths measured for each ROI connectome. Following Definition 4.5, the pruning process ensures that the set $\mathcal{E}_\mathcal{T}$ retains only the significant edges that satisfy the optimization criterion.

### 5.1. Optimal Weighted Tree Path

Given the pruned graph $\mathcal{T}(G)$, we examine paths that integrate node importance and high-order FC across multiple tree depths. Let $P = \{v_0, v_1, \ldots, v_k\}$ represent a path of length $k-1$ in $\mathcal{T}(G)$, where each node $v \in P$ is associated with a scalar score $\mathcal{S}(v; \Theta) \in \mathbb{R}$, computed from Eq. 14. The edge set of $P$ is denoted by $E(P) = \{(v_i, v_j) \mid v_i, v_j \in$

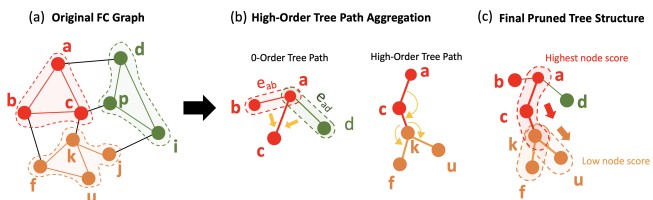

*Figure 2.* **Aggregation for hierarchical neighborhood paths.** Fig. (a) illustrates the original graph structure with weighted connectivity before pruning, providing subnetworks that represent the differences between brain regions. In Fig. (b), the zero-order path is depicted; here, aggregation initiates at the highest-scoring node, which integrates its immediate neighborhood by combining edges $e_{ab}$ and $e_{ad}$, i.e., two direct paths connecting the highest-scoring node to its neighbors. Fig. (c) shows the aggregation of higher-order paths, where isolated nodes are connected along the shortest weighted paths, thereby capturing more complex connectivity patterns beyond immediate neighbors.

$P, (v_i, v_j) \in \mathcal{T}(G)\}$, where each edge $e_{v_i v_j} \in E(P)$ represent the path of neighbor connectivity. The composite path weight is expressed as:

$$\mathcal{W}(P) = \alpha \underbrace{\sum_{v \in P} \mathcal{S}(v; \Theta)}_{\text{Node Score Contribution}} + (1-\alpha) \underbrace{\sum_{s=1}^{S} \sum_{(v_i, v_j) \in E(P)} \mathcal{F}_{v_i v_j}^{(s)}}_{\text{High-Order FC Contribution}}. \tag{16}$$

Here, $\mathcal{F}_{v_i v_j}^{(s)} \in \mathbb{R}$ represents the $s$-th order FC strength between nodes $v_i$ and $v_j$ along the tree path. To solve Eq. 16, we identify the path that minimizes $\mathcal{W}(P)$ by adjusting the parameter $\alpha \in [0, 1]$ to balance the contributions of the scoring function $\mathcal{S}(v; \Theta)$ and the connectivity strengths along the weighted path. Additional experimental results can be found in Fig. 7. According to theorem D.1 in the Appendix, the optimal path can be determined when $\alpha^*$ exist in the interval $[\alpha_L, \alpha_U]$ (Xue, 2000).

### 5.2. High-Order Tree Path Aggregation

In high-order tree path traversal, our objective is to aggregate information of direct and indirect about the connectivity of tree paths within the pruned graph $\mathcal{T}(G)$. In practice, we aggregate connectivity information from neighboring edges at each node, forming a weighted tree path. We define the second term in Eq. 16, which represents the high-order FC contribution for an $s$-order connectivity along the $p$-th path, as follows:

$$\mathcal{F}_{v_i v_j}^{(s)} = \mathcal{F}_{v_i v_j}^{(0)} + \sum_{p \in \mathcal{P}_s(v_i, v_j)} P_s(p) \tag{17}$$

where $\mathcal{F}_{v_i v_j}^{(0)} = \mathcal{F}(v_i) + \mathcal{F}(v_j)$ represents the FC of a 0-th order path, and $\mathcal{P}_s(v_i, v_j)$ denotes the set of all possible $s$-order paths between nodes $i$ and $j$ in $\mathcal{T}(G)$. For

**Algorithm 1** Hierarchical Tree Trunk Extraction with Weighted Tree Path Learning

---
1: **Input:** Node scores $\{\mathcal{S}(v; \boldsymbol{\Theta})\}_{v \in \mathcal{V}}$, pruned graph $\mathcal{T}(G)$ with nodes $\mathcal{V}$ and edges $\mathcal{E}_{\mathcal{T}}$, maximum levels $L_{\max}$, scaling factor $\alpha$.
2: **Output:** Hierarchical trunks $\mathcal{T} = \{\mathcal{T}_1, \mathcal{T}_2, \ldots, \mathcal{T}_{L_{\max}}\}$
3: Initialize $\mathcal{T} \leftarrow \emptyset$.
4: Let $\mathcal{G}(\mathcal{V}, \mathcal{E}_{\mathcal{T}})$ be the pruned graph using MST.
5: **for** $l \leftarrow 1$ to $L_{\max}$ **do**
6:     Identify connected components $\{\mathcal{K}_1, \mathcal{K}_2, \ldots, \mathcal{K}_m\}$ in $\mathcal{G}$.
7:     **for** each component $\mathcal{K}_j$ **do**
8:         Find $v_{\text{start}} = \arg\max_{v \in \mathcal{K}_j} \mathcal{S}(v)$.
9:         Compute shortest paths $\mathcal{P}(v_{\text{start}}, v)$ for all $v \in \mathcal{K}_j$ with weighted path:
$$\mathcal{W}(P) = \alpha \sum_{v \in P} \mathcal{S}(v; \boldsymbol{\Theta}) + (1-\alpha) \sum_{s=1}^{S} \sum_{(v_i, v_j) \in E(P)} \mathcal{F}_{v_i v_j}^{(s)},$$
10:         Let $v_{\text{end}}$ maximize $\mathcal{W}(P)$, and define $P^* = \mathcal{P}(v_{\text{start}}, v_{\text{end}})$.
11:         Append $P^*$ to $\mathcal{T}_l$.
12:     **end for**
13:     Remove edges in $\mathcal{T}_l$ from $\mathcal{G}$ and update the graph.
14:     Combine paths: $\mathcal{T} \leftarrow \mathcal{T} \cup \mathcal{T}_l$.
15: **end for**
16: **Return** $\mathcal{T}$.
---

each $s$-order path index $p = (i, k_1, k_2, \ldots, k_s, j)$, where all intermediate nodes are distinct ($k_m \neq k_n \ \forall m \neq n$), its contribution is calculated as $P_s(p) = \sum_{v \in p} \mathcal{F}_v^{(s)}$. For a path of order $s$, its extended connectivity at order $s+1$-th through the neighboring node $k$ can be computed as $\mathcal{F}_{v_i v_j}^{(s+1)} = \sum_{k \in \mathcal{N}(v_i, v_j)} \left( \mathcal{F}_{v_i v_k}^{(s)} + \mathcal{F}_{v_k v_j}^{(s)} \right)$, where $\mathcal{N}(v_i, v_j)$ represents the set of common neighbors between nodes $i$ and $j$, and $\mathcal{F}_{v_i v_k}^{(s)}$ denotes the $s$-th order path connectivity from node $i$ to node $k$. This formulation allows us to capture direct connections and higher-order structural relationships within the brain network, as illustrated in Fig. 2.

**Fine-Grained Tree Trunk**

We aim to construct a hierarchical tree structure comprising a trunk and branches distributed across different levels in $\mathcal{T}(G)$. Our method groups nodes with similar scores into the same hierarchical level integrates the most relevant paths into the $l$-th trunk $\mathcal{T}_l$ and excludes unrelated paths. For the $l$-th level connected component $G^* = (V_l, E_l)$, the numbers of nodes and edges decrease monotonically as the level increases—that is, $|V_l| \leq |V_{l-1}|$ and $|E_l| \leq |E_{l-1}|$. Following Section 5.1, we identify the shortest path set $P(v_{start}, v_{end}) = \{v_{start}, v_2, \ldots, v_{end}\}$ that incorporates the weighted tree path and the optimal trunk set $\mathcal{T}' = \{\mathcal{T}_1, \mathcal{T}_2, \ldots, \mathcal{T}_k\}$, as presented in algorithm 1.

# 6. Experiments

In this section, we first evaluate the proposed NEUROTREE performance in brain disease classification (section 6.2) and its application to predicting chronological age (section 8). In section 6.4, we analyze the convergence of the spectral norm for $k$-hop connectivity. Finally, we examine how hierarchical brain trees interpret FC patterns between brain regions and their corresponding seven subnetworks in section 7. Detailed experimental settings for NEUROTREE are provided in Appendix H.

## 6.1. Datasets

We validated two publicly available fMRI datasets —one focusing on cannabis use disorder and the other on schizophrenia. ① **Cannabis** (Kulkarni et al., 2023): The cannabis dataset comprises fMRI data from two distinct sources. The data were preprocessed from 3-Tesla fMRI acquisitions, and the mean time series for each subject was computed across 90 ROIs using the Stanford atlas parcellation (Shirer et al., 2012). ② **COBRE** (Calhoun et al., 2012): The Center for Biomedical Research Excellence (COBRE) dataset includes resting-state fMRI data collected from healthy controls and individuals diagnosed with schizophrenia. All MRI data were parcellated into 118 ROI regions using the Harvard-Oxford atlas. We summarized the statistics and demographics of both datasets in Appendix Table 5.

## 6.2. Qualitative Results

This section is dedicated to evaluating the performance of NEUROTREE against state-of-the-art (SOTA) GCN architectures including 1) $Pearson$ GCN 2) $k$-NN GCN 3) GAT (Veličković et al., 2017) 4) BrainGNN (Li et al., 2021) 5) BrainUSL (Zhang et al., 2023) and BrainGSL (Wen et al., 2023a) for graph classification tasks. Additionally, we include the tree path-based model PathNN (Michel et al., 2023). The comparative analysis was conducted using 5-fold cross-validation on two distinct datasets, benchmarking NEUROTREE against both baseline models and SOTA approaches. The comprehensive experimental results are presented in Table 1.

**Compare with Brain GCN Models.** We quantitatively assessed brain network construction using various metrics, employing Pearson correlation and KNN as connectivity measures. As shown in Table 1, the GAT model effectively integrates attention mechanisms into fMRI node features, achieving mean AUC scores of 0.72 and 0.67. In contrast, the conventional GCN is limited to learning solely from topological structures. Furthermore, NEUROTREE enhances ACC performance in fMRI-based models, improving BrainGNN (ACC ↑ 23.72%), BrainUSL (ACC ↑ 12.30%), and BrainGSL (ACC ↑ 12.30%) on the cannabis dataset. On the COBRE dataset, BrainGNN, BrainUSL, and BrainGSL

*Table 1.* Evaluating graph classification performance with five-fold cross-validation. We computed the most competitive baseline for each method. We compared the second-best methods denoted by blue color and calculated the improvement rate, denoted as "Improv. (%)".

| | Cannabis | | | | COBRE | | | |
|---|---|---|---|---|---|---|---|---|
| **Model** | **AUC** | **Acc.** | **Prec.** | **Rec.** | **AUC** | **Acc.** | **Prec.** | **Rec.** |
| $Pearson$ GCN | 0.67±0.06 | 0.55±0.07 | 0.59±0.13 | 0.55±0.06 | 0.54±0.11 | 0.55±0.10 | 0.61±0.12 | 0.55±0.10 |
| $k$-NN GCN | 0.64±0.03 | 0.62±0.03 | 0.63±0.03 | 0.63±0.03 | 0.66±0.07 | 0.62±0.08 | 0.63±0.08 | 0.63±0.08 |
| GAT (Veličković et al., 2017) | 0.72±0.05 | 0.67±0.04 | 0.70±0.06 | 0.67±0.04 | 0.67±0.08 | 0.60±0.11 | 0.57±0.21 | 0.60±0.11 |
| BrainGNN (Li et al., 2021) | 0.67±0.13 | 0.59±0.16 | 0.51±0.28 | 0.59±0.12 | 0.55±0.11 | 0.50±0.02 | 0.31±0.11 | 0.50±0.02 |
| BrainUSL (Zhang et al., 2023) | 0.63±0.11 | 0.65±0.06 | 0.62±0.13 | 0.63±0.11 | 0.57±0.10 | 0.54±0.04 | 0.41±0.18 | 0.57±0.11 |
| BrainGSL (Wen et al., 2023a) | 0.59±0.11 | 0.65±0.02 | 0.67±0.17 | 0.65±0.02 | 0.55±0.12 | 0.51±0.04 | 0.45±0.11 | 0.51±0.04 |
| MixHop (Abu-El-Haija et al., 2019) | 0.73±0.05 | 0.69±0.03 | 0.70±0.04 | 0.69±0.03 | 0.69±0.05 | 0.61±0.06 | 0.62±0.07 | 0.61±0.06 |
| GPC-GCN (Li et al., 2022b) | 0.53±0.05 | 0.60±0.06 | 0.37±0.08 | 0.60±0.06 | 0.50±0.00 | 0.47±0.04 | 0.22±0.04 | 0.47±0.04 |
| PathNN (Michel et al., 2023) | 0.70±0.10 | 0.67±0.04 | 0.72±0.12 | **0.83**±0.16 | 0.49±0.01 | 0.51±0.05 | 0.32±0.27 | 0.43±0.46 |
| Ours (w/o $\theta$) | 0.49±0.01 | 0.60±0.06 | 0.37±0.08 | 0.60±0.06 | 0.50±0.00 | 0.47±0.04 | 0.22±0.01 | 0.47±0.04 |
| Ours (w/o $\mathcal{L}_{CMFC}$ ) | 0.74±0.08 | 0.73±0.05 | 0.73±0.04 | 0.73±0.05 | 0.69±0.10 | 0.63±0.10 | 0.64±0.10 | 0.63±0.10 |
| NEUROTREE | **0.80**±0.05 | **0.73**±0.04 | **0.73**±0.04 | 0.74±0.04 | **0.71**±0.10 | **0.65**±0.08 | **0.66**±0.08 | **0.65**±0.08 |
| Improv. (%) | 8.11% | - | - | 1.37% | 2.89% | 3.17% | 3.12% | 3.17% |

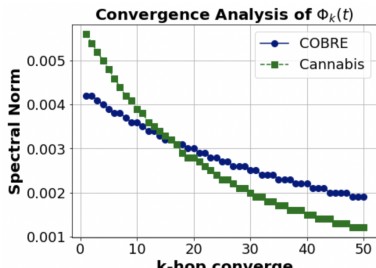 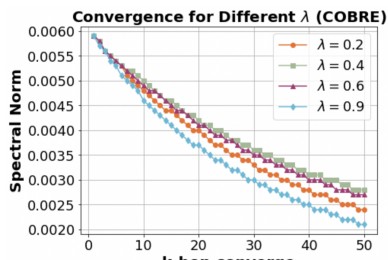 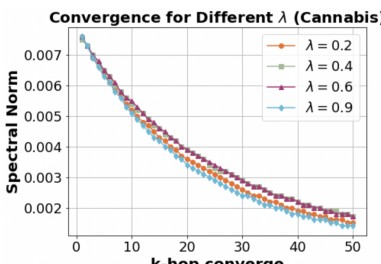

*Figure 3.* **Convergence analysis of $\Phi_k(t)$ over $k$-hop** The spectral norm of $\Phi_k(t)$ reveals differential convergence rates across varying $k$-orders among distinct mental disorders, notably demonstrating that cannabis exhibits a steeper convergence gradient compared to COBRE as $\lambda$ increases.

further improve ACC by 30.00%, 20.37%, and 27.45%, respectively.

**Compare with Path-Based Models.** As shown in Table 1, NEUROTREE outperforms existing high-connectivity view field-aware graph GCN models, such as the high-order concatenation layer-based MixHop, achieving over 6% average performance improvement in AUC across all metrics on the cannabis dataset. Furthermore, compared to MixHop on the COBRE dataset, NEUROTREE not only achieved a 2% increase in AUC, but also demonstrated a mean performance improvement of 6.3% to 6.5% in ACC, Precision, and Recall. Notably, although GPC-GCN can aggregate multi-level connectivity information by considering the concept of paths, it does not significantly improve classification performance on critical connectivity paths. Similarly, the PathNN framework aggregates path information starting from nodes to generate node representations but shows only limited improvements. In contrast, our NEUROTREE achieves a 14% improvement in AUC on the cannabis dataset, while PathNN underperforms compared to MixHop on the COBRE dataset. The experimental results indicate that NEUROTREE is superior in capturing temporal FC patterns and predicting outcomes in addiction and psychiatric disorders.

### 6.3. Ablation Study

**The Impact of Objective Function and Demographics.** In our experiments, we evaluate the efficacy of CMFC loss integration within our proposed NEUROTREE. The quantitative analysis presented in Table 1 indicates modest improvements in the cannabis dataset; however, its implementation in the COBRE dataset achieved an AUC of 0.71, with consistent performance gains exceeding 3% across metrics, particularly in the characterization of mental disorder features. This enhancement stems from combining the loss function with CMFC loss optimization framework that simultaneously suppresses connectivity between dissimilar regions while amplifying patterns among similar neurological areas. Furthermore, our ablation experiments demonstrate that age serves as a critical modulator in AGE-GCN by adapting parameter $\theta$, substantially influencing the classification of changes in brain connectivity.

### 6.4. Convergence Analysis

The intuition for analyzing the rate of spectral norm decay with increasing $k$-hop allows us to examine how rapidly information from distant nodes attenuates across the dynamic brain network structure. More detalil explanation

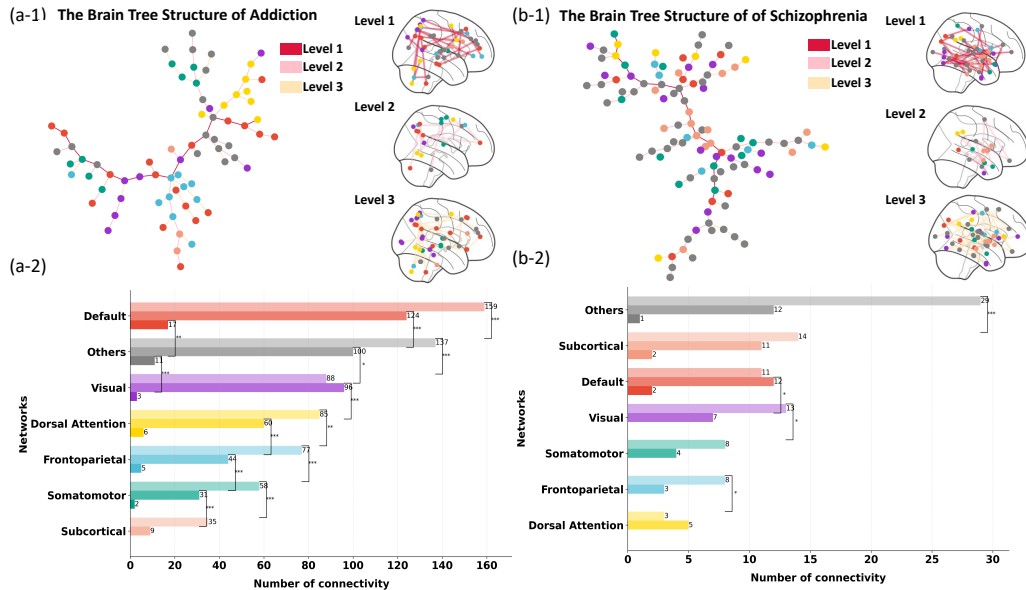

*Figure 4.* The visualization of the brain tree illustrates psychiatric disorders structured into three hierarchical trunk levels. Panels (a-1) and (b-1) mark the most significant nodes along the tree path. The *l*-1 pathways represent regional connectivity, corresponding to the level three brain maps on the right. Panels (a-2) and (b-2) depict the number of connections using color gradients across the hierarchical levels.

can be found in Appendix I. To further examine the convergence speed of $k$-hop FC in various brain disorders, we analyze the spectral norm under the $k = 1$ setting for the convergence of $\Phi_k(t)$ in Fig. 3 (left). Although the COBRE dataset initially exhibits slower convergence, particularly with the boundary defined around order 15, the overall convergence in the cannabis dataset is faster than in COBRE across different $k$-hop values. Furthermore, when analyzing the convergence of $\Phi_k(t)$ under varying $\lambda$ values, as shown in Fig. 3 (middle) and (right), the $\lambda$ values for cannabis are more concentrated and stable. In contrast, COBRE requires larger $\lambda$ values to achieve convergence. The empirical findings demonstrate that examining the spectral norm of $\Phi_k(t)$ across various $k$ values effectively reveals the relationship between high-order patterns and decay characteristics of FC in different mental disorders.

## 7. Brain Tree Analysis

This section investigates brain disorders by analyzing hierarchical brain networks using decoded fMRI-derived connectomes. NEUROTREE deconstructs the fMRI brain network into a tree-structured component graph consisting of a main trunk and radial branches. We identify key regions associated with brain disorders within each trunk by analyzing optimal weighted high-order tree paths. Our analysis highlights the most relevant functional network regions determining critical pathways within the brain connectome tree.

**Exploring Hierarchical Regional Patterns in Brain Disorders.** To better understand how embedded brain features

vary across different hierarchical levels of brain subnetworks, NEUROTREE predicts each ROI score and performs reranking, as illustrated in Fig. 1. Higher levels represent trunk pathways of high-scoring nodes, strongly connected edges, and their associated brain ROIs. We mapped the atlas parcellations of both datasets (i.e., 90 ROIs and 118 ROIs) to Yeo's seven-network parcellation (Yeo et al., 2011), which includes the Visual Network (VN), Somatomotor Network (SMN), Dorsal Attention Network (DAN), Ventral Attention Network (VAN), Frontoparietal Network (FPN), and Default Mode Network (DMN). Additionally, we incorporated the Subcortical System (SUB) in our analysis to examine the subcortical structures. Atlas regions not belonging to these networks were classified as 'Others.' More details can be found at Appendix L.2.

Next, we discuss the addiction brain tree illustrated in Fig. 4 (a-1) and (a-2), which features three trunk levels. At the primary trunk level (red pathway), the addiction brain tree predominantly exhibits connectivity among DMN (red) and VN (purple) nodes; similarly, in Fig. 4 (b-1) and (b-2), the schizophrenia brain tree at the main trunk level primarily involves SUB (pink) nodes and partial VN nodes. At the second hierarchical level (pink pathways), the addiction cohort exhibits predominant connections within DAN, DMN, and VN nodes. In contrast, the schizophrenia group at this level demonstrates connectivity primarily among VN nodes and some FPN (light blue) nodes. At the third hierarchical level (yellow pathways), the terminal branches in the addiction group show minimal connections with the SMN nodes (green) and DAN nodes, whereas the schizophrenia cohort displays several connections among DMN nodes.

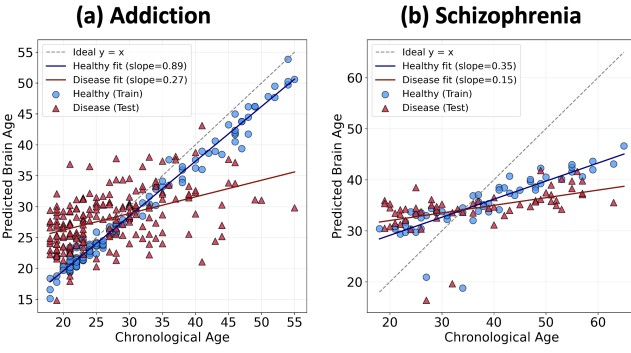

*Figure 5.* The scatterplot shows the gaps between fMRI-predicted brain age and chronological age for healthy control and mental disorder groups.

## 8. Brain Age Estimation

Comparing estimated fMRI brain age against actual chronological age in individuals with mental disorders can provide valuable insights into the severity and progression of these conditions (Stankevičiūtė et al., 2020). The physiological actual age of the human brain is unknown and cannot be directly measured; however, we can predict brain age through regression tasks on fMRI data using NeuroTree learned fMRI graph embeddings. We hypothesize that there exist certain gaps between the actual chronological age of healthy individuals and the brain age of those with mental disorders. For example, when the gap between model-predicted brain age and actual chronological age is positive, it indicates accelerated brain aging.

To validate model prediction in brain age, we trained NeuroTree on a healthy control cohort and predicted non-healthy subjects to better measure brain age gaps. In Table 2, we demonstrate NeuroTree's predictive performance across different age groups in two datasets. Notably, the younger age group (18-25 years) exhibited markedly lower Mean Squared Error (MSE) values of 9.51 in the Cannabis dataset and 4.48 in the COBRE dataset compared to other age groups, indicating greater accuracy in brain age prediction for younger individuals. Compared to younger age groups, NeuroTree's prediction MSE across both mental disorder datasets provides valuable insight, showing that brain age estimation for older individuals may be affected by greater individual variability in brain changes associated with mental disorders after middle age, while younger groups likely exhibit more consistent patterns of FC during neurodevelopment.

## 9. Discussion

In section 7, our analysis of the tree-structured topology revealed distinct patterns of brain subnetwork alterations associated with addiction and schizophrenia. These changes

*Table 2.* NEUROTREE prediction for different age groups.

| Datasets | Age Group | MSE | Pearson (r) |
|---|---|---|---|
| Cannabis | 18-25 (n=150) | $9.51_{\pm 1.67}$ | $0.27_{\pm 0.02}$ |
| | 25-40 (n=127) | $27.32_{\pm 3.15}$ | $0.18_{\pm 0.01}$ |
| | 40-54 (n=41) | $25.12_{\pm 1.40}$ | $0.18_{\pm 0.00}$ |
| COBRE | 18-25 (n=35) | $4.48_{\pm 0.42}$ | - |
| | 25-40 (n=56) | $37.64_{\pm 2.02}$ | $0.10_{\pm 0.01}$ |
| | 40-66 (n=61) | $32.64_{\pm 3.43}$ | $0.49_{\pm 0.13}$ |

were primarily concentrated in networks governing cognitive functions, including the DMN, the visual task VN and the decision-making FPN. This network-specific disruption pattern aligns with previous findings (Kleinhans et al., 2020; Kulkarni et al., 2023; Ding et al., 2024b).

Prior studies have established disrupted connectivity patterns in schizophrenia, particularly within the DMN (Ishida et al., 2024; Yang et al., 2022). We identified intermittent abnormal coupling between subcortical regions, especially the connections with the prefrontal cortex and cerebellum (Ding et al., 2024a; Hancock et al., 2023). Furthermore, areas of the visual cortex were correlated with impaired motor coordination and sensorimotor integration (Verma et al., 2023; Zhao et al., 2022), consistent with the high FC regions illustrated in Fig. 4. Additionally, our analysis revealed distinct age-related differences in brain FC between addiction and schizophrenia (Table 2). We further verify our finding in Fig. 5, which illustrates the relationship between predicted brain age and chronological age. The predicted brain age for both mental disorders is lower than the chronological age, particularly in the higher age group. Both conditions appear to affect age-related brain characteristics, leading to an abnormal relationship between brain age and chronological age. Notably, addiction demonstrates a more significant impact on brain development and aging than schizophrenia, which is consistent with our observations in section 8.

## 10. Conclusion

This study proposed NEUROTREE, a novel hierarchical brain tree framework that integrates $k$-hop AGE-GCN with neural ODEs and a contrastive mask FC loss to decode functional brain pathways in psychiatric disorders. NEUROTREE effectively incorporates demographic to improve brain disease classification and enhances interpretability in predicting brain age while revealing distinct hierarchical connectivity patterns within the DMN, FPN, and VN across different mental disorders. Additionally, these findings demonstrate that different subtypes of mental disorders exhibit varying trajectories of FC changes across age periods. These insights contribute to the neuroscientific understanding of functional alterations induced by mental disorders, potentially informing therapeutic interventions and pharmacological treatments.

## Impact Statement

In this study, we examine the increasing societal impact of mental illness and addiction, highlighting the critical importance of understanding the underlying mechanisms of brain disorders. Traditional GNNs designed for fMRI data have proven effective in capturing network-based features; however, many of these approaches lack interpretability and fail to incorporate external demographic information within a deep learning framework. To address these limitations, we introduce NEUROTREE, a novel framework that establishes a robust foundation for advancing neuroscience research, identifies influential brain network regions associated with disorders, and elucidates potential therapeutic targets for future drug development and intervention strategies in specific brain regions. In future work, our NeurTree framework can be extended to related research fields such as EEG and social networks.

## Reproducibility

All of our experiments are reproducible, with results presented in Table 1 and Table 2. We will upload the preprocessed data to Google Drive and release our code on GitHub upon acceptance of the paper. Please note that, based on our experience, there may be some variations in the results shown in Table 1 and Table 2 due to differing GPU configurations. For accurate results, please refer to our released checkpoints.

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

## A. Appendix: Proof Formula 7 for K-hop Connectivity

**Theorem A.1** (K-hop Connectivity). *Given the k-hop connectivity adjacency operator:*

$$\hat{A}_k(t) = \Gamma \odot A^s \odot \left[\lambda A^d(t) + (1 - \lambda)(A^d(t))^T\right]^k, \tag{18}$$

*the k-hop connectivity can be expressed as:*

$$\left[\lambda A^d(t) + (1 - \lambda)(A^d(t))^T\right]^k = \sum_{i=0}^{k} \binom{k}{i} \lambda^i (1 - \lambda)^{k-i} \left(A^d(t)\right)^i \left((A^d(t))^T\right)^{k-i}. \tag{19}$$

*Proof.* For $k = 1$, the formula holds trivially:

$$\left[\lambda A^d(t) + (1 - \lambda)(A^d(t))^T\right]^1 = \lambda A^d(t) + (1 - \lambda)(A^d(t))^T.$$

Assume the formula holds for $k = n$:

$$\left[\lambda A^d(t) + (1 - \lambda)(A^d(t))^T\right]^n = \sum_{i=0}^{n} \binom{n}{i} \lambda^i (1 - \lambda)^{n-i} \left(A^d(t)\right)^i \left((A^d(t))^T\right)^{n-i}.$$

For $k = n + 1$, expand:

$$\left[\lambda A^d(t) + (1 - \lambda)(A^d(t))^T\right]^{n+1} = \left[\lambda A^d(t) + (1 - \lambda)(A^d(t))^T\right]^n \left[\lambda A^d(t) + (1 - \lambda)(A^d(t))^T\right].$$

Substituting the inductive hypothesis and applying Pascal's identity:

$$\binom{n+1}{i} = \binom{n}{i-1} + \binom{n}{i},$$

We obtain:

$$\left[\lambda A^d(t) + (1 - \lambda)(A^d(t))^T\right]^{n+1} = \sum_{i=0}^{n+1} \binom{n+1}{i} \lambda^i (1 - \lambda)^{n+1-i} \left(A^d(t)\right)^i \left((A^d(t))^T\right)^{n+1-i}.$$

Thus, by induction, the formula holds for all $k$. And we have following corollary in A.2 $\qquad\square$

**Corollary A.2** (Asymmetric Property of $K$-hop Operator). *The k-hop operator preserves directional information:*

$$\hat{A}_k(t) \neq (\hat{A}_k(t))^T, \forall \lambda \neq \frac{1}{2} \tag{20}$$

*Proof.* By the definition of the $k$-hop connectivity operator:

$$\hat{A}_k(t) = \gamma \odot A^s \odot \left[\lambda A^d(t) + (1 - \lambda)(A^d(t))^T\right]^k \tag{21}$$

Applying the binomial expansion from A.1:

$$\hat{A}_k(t) = \Gamma \odot A^s \odot \sum_{i=0}^{k} \binom{k}{i} \lambda^i (1 - \lambda)^{k-i} (A^d(t))^i ((A^d(t))^T)^{k-i} \tag{22}$$

Taking the transpose of $\hat{A}_k(t)$:

$$(\hat{A}_k(t))^T = \Gamma^T \odot (A^s)^T \odot \sum_{i=0}^{k} \binom{k}{i} \lambda^i (1 - \lambda)^{k-i} ((A^d(t))^i)^T (((A^d(t))^T)^{k-i})^T$$

$$= \Gamma \odot A^s \odot \sum_{i=0}^{k} \binom{k}{i} \lambda^i (1 - \lambda)^{k-i} ((A^d(t))^T)^i (A^d(t))^{k-i} \tag{23}$$

When $\lambda \neq \frac{1}{2}$, the weights $\lambda^i$ and $(1-\lambda)^{k-i}$ are asymmetric, meaning:

$$\lambda^i(1-\lambda)^{k-i} \neq \lambda^{k-i}(1-\lambda)^i \tag{24}$$

Therefore, $\hat{A}_k(t) \neq (\hat{A}_k(t))^T$ unless all terms in the expansion are zero. $\qquad\square$

## B. Appendix: Proof of Theorem 4.2

We prove the $\|\hat{A}_k(t)\|_2$ has an upper bound as $k$-hop approaches infinity:

$$\lim_{k \to \infty} \|\hat{A}_k(t)\|_2 \leq \|\Gamma\|_2 \cdot \|A^s\|_2 \cdot \max(\lambda, 1-\lambda)^k \tag{25}$$

*Proof.* Using matrix norm properties:

$$\|A \odot B\|_2 \leq \|A\|_2 \cdot \|B\|_2 \tag{26}$$

Applying the triangle inequality when $\lambda \in [0,1]$, we obtain:

$$\|\lambda A + (1-\lambda)B\|_2 \leq \max(\lambda, 1-\lambda)(\|A\|_2 + \|B\|_2) \tag{27}$$

Applying the $l^2$-norm to $\hat{A}_k(t)$:

$$\|\hat{A}_k(t)\|_2 = \|\Gamma \odot A^s \odot [\lambda A^d(t) + (1-\lambda)(A^d(t))^T]^k\|_2 \tag{28}$$

And we have:

$$\|\hat{A}_k(t)\|_2 \leq \|\Gamma\|_2 \cdot \|A^s\|_2 \cdot \|[\lambda A^d(t) + (1-\lambda)(A^d(t))^T]^k\|_2 \tag{29}$$

Using Eq. 27, we further bound:

$$\|[\lambda A^d(t) + (1-\lambda)(A^d(t))^T]\|_2 \leq \max(\lambda, 1-\lambda)(\|A^d(t)\|_2 + \|(A^d(t))^T\|_2) \tag{30}$$

Since $\|(A^d(t))^T\|_2 = \|A^d(t)\|_2$, we obtain:

$$\|[\lambda A^d(t) + (1-\lambda)(A^d(t))^T]\|_2 \leq 2\max(\lambda, 1-\lambda)\|A^d(t)\|_2 \tag{31}$$

Taking the power of $k$ on both sides of Eq. 31, we have $\|\hat{A}_k(t)\|_2 \leq \|\hat{A}(t)\|_2^k$ and the following inequality holds:

$$\|[\lambda A^d(t) + (1-\lambda)(A^d(t))^T]^k\|_2 \leq \max(\lambda, 1-\lambda)^k \tag{32}$$

Substituting into equation 29, we get:

$$\|\hat{A}_k(t)\|_2 \leq \|\Gamma\|_2 \cdot \|A^s\|_2 \cdot \max(\lambda, 1-\lambda)^k. \tag{33}$$

Since $\max(\lambda, 1-\lambda) < 1$ for $\lambda \in (0,1)$, the series converges geometrically. This result indicates that $\|\hat{A}_k(t)\|$ converges to zero as $k \to \infty$, provided the spectral radius of $A$, denoted by $\lambda$, satisfies $\max(\lambda, 1-\lambda) < 1$. Consequently, the rate of decay of $\|\hat{A}_k(t)\|$ is governed by $\max(\lambda, 1-\lambda)^k$, which demonstrates the dependence on the spectral properties of $A$ and ensures convergence for sufficiently small $\lambda$. $\qquad\square$

## C. Appendix: Proof for Proposition 4.3

To prove the proposition 4.3, we express the neural ODE system as:

$$\frac{dH^{(l)}(t)}{dt} = F(H^{(l)}(t), t) \tag{34}$$

where

$$F(H^{(l)}(t), t) = \sigma\left(\sum_{k=0}^{K-1} \Phi_k(t) H^{(l)}(t) W_k^{(l)}\right) \tag{35}$$

*Proof.* Assume $\sigma$ is a differentiable function and continuous in $t$. We show that $F$ satisfies Lipschitz continuity in $H^{(l)}$ and continuity in $t$. Let $H_1^{(l)}$ and $H_2^{(l)}$ be two output states:

$$
\begin{aligned}
&\|F(H_1^{(l)}, t) - F(H_2^{(l)}, t)\|_2 \\
&= \left\|\sigma\left(\sum_{k=0}^{K-1} \Phi_k(t) H_1^{(l)} W_k^{(l)}\right) - \sigma\left(\sum_{k=0}^{K-1} \Phi_k(t) H_2^{(l)} W_k^{(l)}\right)\right\|_2
\end{aligned} \tag{36}
$$

since $\|F(H_1^{(l)}, t) - F(H_2^{(l)}, t)\|_2$ is bounded by the Lipschitz constant of $\sigma$, denoted by $L_\sigma$, we obtain:

$$\|F(H_1^{(l)}, t) - F(H_2^{(l)}, t)\|_2 \leq L_\sigma \left\|\sum_{k=0}^{K-1} \Phi_k(t)(H_1^{(l)} - H_2^{(l)}) W_k^{(l)}\right\|_2 \tag{37}$$

Using matrix norm properties and $\|\Phi_k(t)\|_2 \leq 1$ and $W_k^{(l)}$ are bounding the norms. The bound equation can be written as:

$$\|F(H_1^{(l)}, t) - F(H_2^{(l)}, t)\|_2 \leq L_\sigma \sum_{k=0}^{K-1} \|W_k^{(l)}\|_2 \|H_1^{(l)} - H_2^{(l)}\|_2 \tag{38}$$

Let $M = \max_k \|W_k^{(l)}\|_2$. Then:

$$\|F(H_1^{(l)}, t) - F(H_2^{(l)}, t)\|_2 \leq L_\sigma KM \|H_1^{(l)} - H_2^{(l)}\|_2 \tag{39}$$

Thus, $F$ is Lipschitz continuous in $H^{(l)}$ with Lipschitz constant $L_F = L_\sigma KM$. For continuity in $t$, we use the Lipschitz condition of $\Phi_k(t)$:

$$\|\Phi_k(t_1) - \Phi_k(t_2)\|_2 \leq L \|t_1 - t_2\|_2 \tag{40}$$

Thus, Eq. 40 ensures that $F$ is continuous in $t$. According to Picard's existence theorem, the $k$-hop ODE-GCN has a unique solution. $\qquad \square$

## D. Appendix: Finding the Shortest Paths in a Weighted Brain Tree

**Theorem D.1** (Optimal Path Extremes). *Let $\mathcal{W}(S) = \sum_{v \in P} S(v; \Theta)$ represent the node score contribution and $\mathcal{W}(C) = \sum_{e_{ij} \in E(P)} \mathcal{C}_{i,j}$ represent the connectivity strength contribution in the pruned graph $\mathcal{T}(G)$. For an arbitrary integer threshold $\lambda^*$, the optimal path $P^*$ satisfies the following interval constraint $\mathcal{W}(S) \leq \lambda^* \leq \mathcal{W}(C)$.*

In practice, the minimum interval can be determined using the bisection method within $[\alpha_L, \alpha_U]$ using a bisection method such that $\mathcal{W}(S) \leq \mathcal{W}_{\alpha_L}(P^*) \leq \alpha^* \leq \mathcal{W}_{\alpha_U}(P^*) \leq \mathcal{W}(C)$ (Xue, 2000).

## E. Prediction of FC Alterations in Brain Age Across Different Age Groups

This section presents our application of NEUROTREE for predicting mental disorders and examining their impact on brain FC changes across various age cohorts. We utilized AGE-GCN embeddings to reconstruct FC values for different age groups. Subsequently, we analyzed FC values for various age groups, as illustrated in Fig. 6. The results presented in Fig. 6 (a) and (b) reveal that significant within-group differences were observed exclusively among cannabis users in the 18–25 age group. We observe significant changes in FC values associated with cannabis use disorder and schizophrenia across various age groups, indicating age-dependent variations in FC patterns among individuals with these psychiatric disorders.

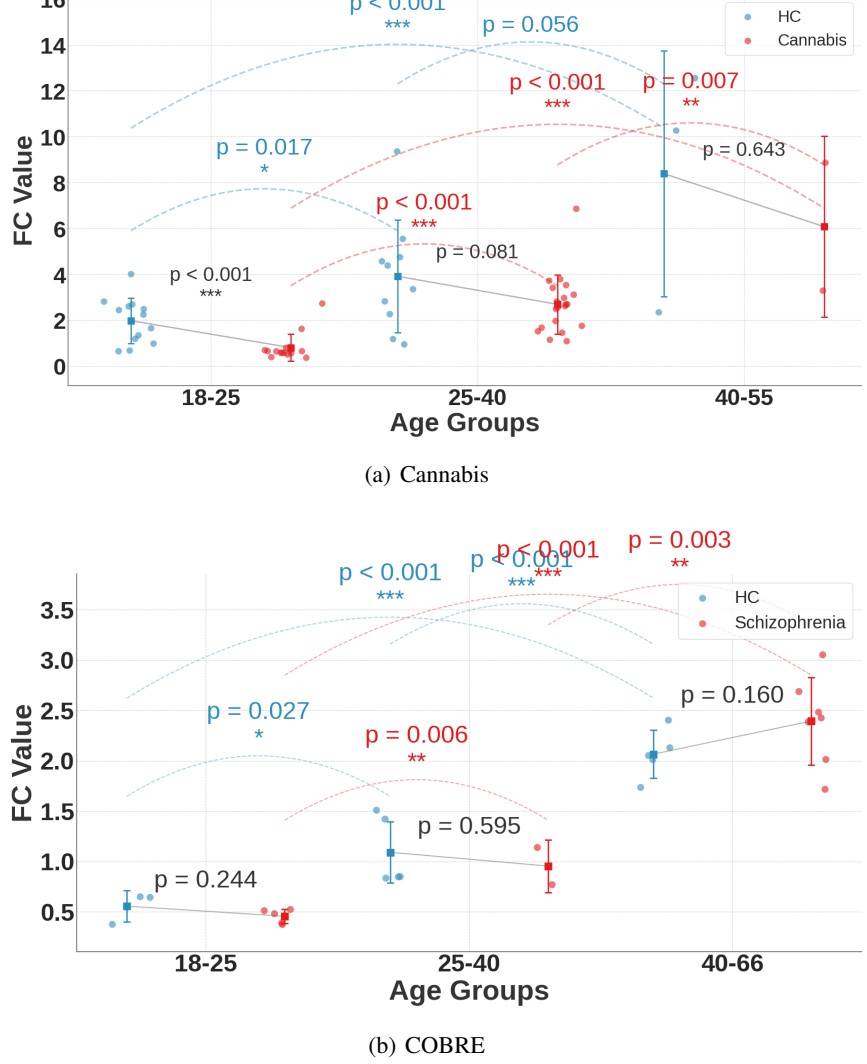

(a) Cannabis

(b) COBRE

Figure 6. Age-related changes in predicted FC values across different groups. (a) cannabis users versus healthy controls (HC) and (b) schizophrenia patients versus HC using the COBRE dataset. Note that asterisks indicate statistical significance ($^{*}p < 0.05$, $^{**}p < 0.01$, $^{***}p < 0.001$). Error bars represent standard error of the mean.

## F. High-Order Tree Path Influence

In further examining the characterization of fMRI in the tree path, we investigated the contribution of FC strength through Eq. 16 across two distinct subtypes of fMRI representations, as illustrated in Fig. 7 Our observations revealed that cannabis-related pathways exhibited relatively higher average FC weights compared to the COBRE dataset. Notably, both the addiction and schizophrenia groups demonstrated significant decreases following an increase in $\alpha$, while the HC cohort maintained relatively stable patterns. These findings provide compelling evidence that higher-order brain tree paths effectively capture the differential contributions of FC strength arising from alterations related to addiction or schizophrenia.

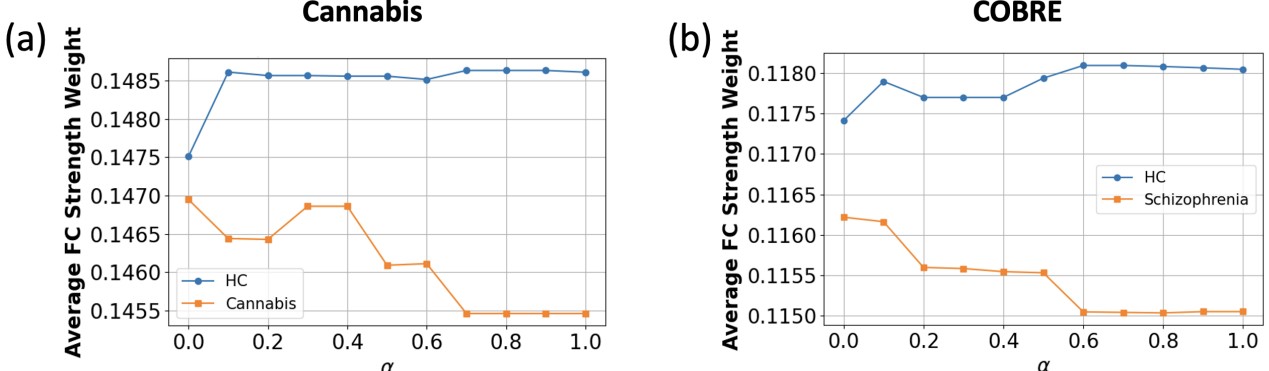

*Figure 7.* Average FC value weights versus $\alpha$ in high-order tree paths, comparing (a) HC vs cannabis users and (b) HC versus schizophrenia patients. The results indicate reduced connectivity patterns with increasing $\alpha$ in both clinical groups compared to the control group.

## G. More Related Work

Schizophrenia (SCZ) and substance abuse (SA) both profoundly impact brain structure and function, with a particularly high comorbidity rate observed among younger populations. Research indicates that SCZ patients frequently exhibit functional abnormalities in critical brain regions, including the DMN, central executive network (CEN), and visual pathways (Ma et al., 2023). However, findings remain inconsistent regarding whether comorbid SA history in SCZ patients exacerbates these neural impairments. (Wojtalik & Barch, 2014) conducted fMRI scans during working memory tasks and discovered significantly enhanced activation in frontal-parietal-thalamic regions among control subjects with SA history. Interestingly, SCZ patients showed no significant activation differences regardless of SA history, suggesting that pre-existing neural deficits in SCZ may obscure the additional effects of SA (Wojtalik & Barch, 2014). In addition, (Passiatore et al., 2023) further demonstrated that FC alterations associated with SCZ risk emerge during adolescence, particularly in cerebellum-occipitoparietal and prefrontal-sensorimotor regions, strongly correlating with polygenic risk scores (PRS).

Beyond network abnormalities, SCZ and SA patients commonly exhibit accelerated brain aging, where their brain-predicted age significantly exceeds their chronological age, indicating heightened functional deterioration of the brain—a phenomenon with critical implications for understanding cognitive decline and disease risk (Cole & Franke, 2017; Lombardi et al., 2021). With the emergence of GNN applications to fMRI data, researchers can now model and analyze brain regions as nodes and functional connections as edges in graph structures. The Pooling Regularized GNN (PR-GNN) proposed by (Li et al., 2020). Applying this methodology to SCZ with comorbid SA may simultaneously facilitate classification and biomarker identification, while incorporating demographic variables such as age, gender, and SA history enables individualized modeling. In summary, SCZ and SA have age-related and highly intertwined effects on brain functional networks. GNN methodologies provide technical tools with both explanatory and predictive capabilities, offering promise for deepening our understanding of the neural mechanisms underlying psychiatric disorders and their potential clinical applications.

## H. Experiments Detail

**Experiments Setting**

**Baseline Models.**  The classical methods for constructing brain networks include Pearson correlation and $k$-nearest neighbor approaches combined with GCN, such as:These methods are as follows:1) $Pearson$ GCN: This model constructs an fMRI graph using Pearson correlation coefficients to represent relationships between ROIs. 2) $k$-NN GCN: This model builds brain networks by connecting $k$-nearest neighbors based on similar input features within brain regions in the graph. 3) GAT (Veličković et al., 2017): This model extends the GCN framework by incorporating an attention mechanism that enhances the aggregation and learning of features from neighboring nodes. 4) BrainGNN (Li et al., 2021): BrainGNN is an interpretable GNN framework designed for analyzing fMRI data. This framework effectively identifies and characterizes significant brain regions and community patterns associated with specific neurological conditions. 5) BrainUSL (Zhang et al., 2023): BrainUSL learns the graph structure directly from BOLD signals while incorporating sparsity constraints and contrastive learning to capture meaningful connections between brain regions. Finally, we compare NEUROTREE with other methods that incorporate higher-order node information, specifically MixHop (Abu-El-Haija et al., 2019) and graph path-based models such as GPC-GCN (Li et al., 2022b) and PathNN (Michel et al., 2023).

**Evaluation Metrics**  In our fMRI graph classification experiments (Section 6.2), we evaluate NEUROTREE using Area Under the Curve (AUC), Accuracy (Acc.), Precision (Prec.), and Recall (Rec.). In brain age prediction (Section 8), we evaluate NEUROTREE predicted age and chronological age using MSE metric.

**Parameters Setting**  In this study, two datasets were trained with a batch size of 16 for 100 epochs using a learning rate of 0.001.

*Table 3.* Training Parameters.

| Notation | Meaning | Value |
|---|---|---|
| $\rho$ | Scale parameter | 0.5 |
| $T$ | Number of time segment | 2 |
| $\lambda$ | $k$-hop connectivity balance between dynamic FC matrices $A^d(t)$ | (0,1) |
| $\beta$ | Learnable age-modulated parameter | (0,1) |
| $\Gamma$ | Learnable graph weight parameter | - |
| $W_k^{(l)}$ | Learnable weight matrix in $l$th layer | - |

**Additional Ablation Study**  In this subsection, we conduct ablation experiments on the performance of brain networks classification with respect to two parameters $\lambda$ and $\Gamma$ on two mental disorders dataset. The experimental results demonstrate that for Cannabis, the model achieves optimal accuracy (0.8) and AUC value (0.87) at $\lambda = 0.2$, with performance decreasing as $\lambda$ increases to 0.4, followed by a recovery at $\lambda = 0.6$. In contrast, COBRE exhibits a different parameter sensitivity pattern, with accuracy showing a decreasing-then-increasing trend as $\lambda$ increases from 0.2 to 0.8, ultimately reaching peak accuracy (0.8) at $\lambda = 0.8$. Regarding the $\Gamma$ parameter, Cannabis shows an overall decline in accuracy as $\Gamma$ increases. On the other hand, COBRE reaches its optimal accuracy (0.8) and AUC value (0.79) at $\Gamma = 0.6$, but experiences a sharp performance drop at $\Gamma = 0.8$. These findings suggest that Cannabis-related brain network structures are more effectively captured at lower $\lambda$ values, whereas COBRE-related brain functional abnormalities are more efficiently identified at higher $\lambda$ and moderate $\Gamma$ values.

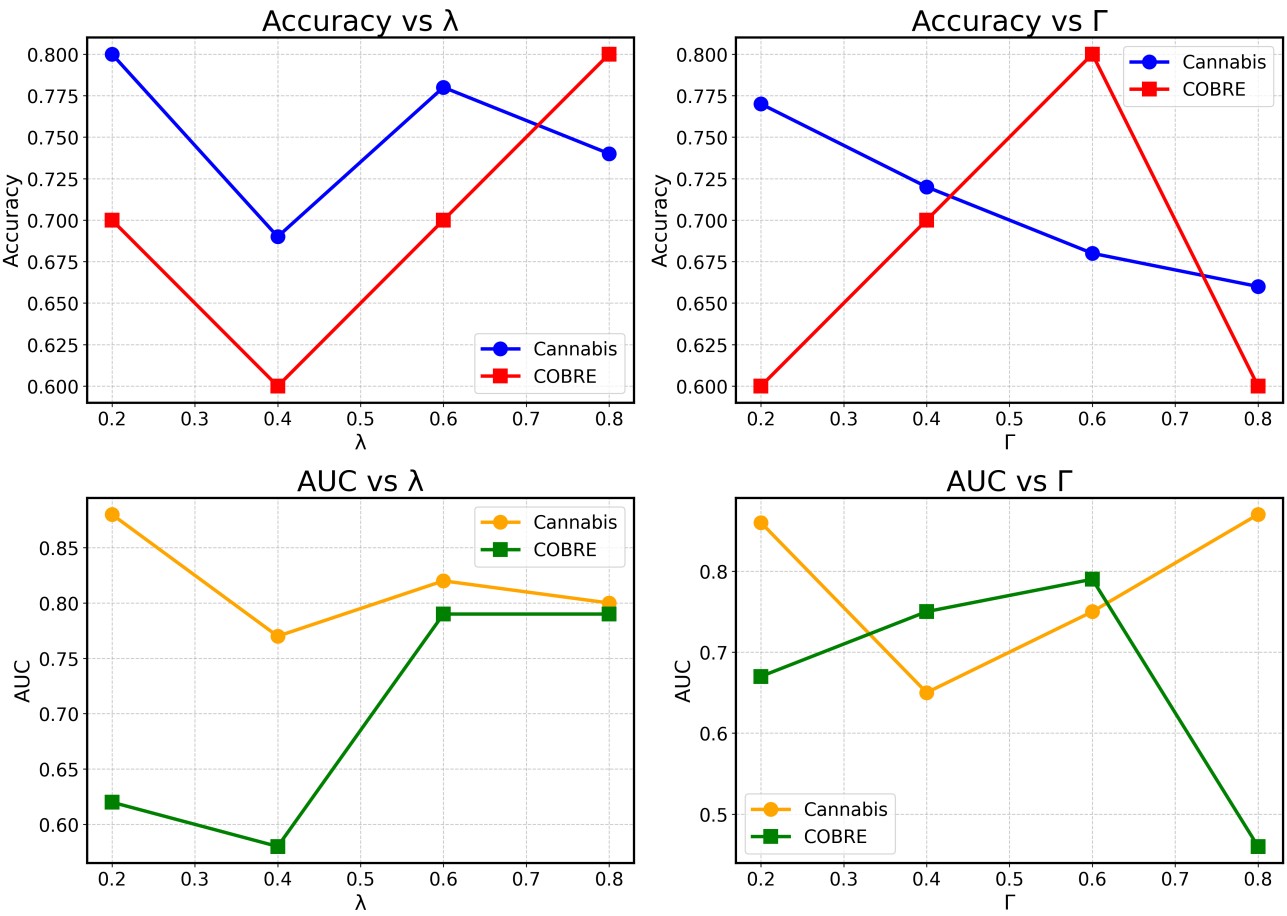

*Figure 8.* Evaluation of brain network classification performance based on model parameters $\lambda$ and $\Gamma$.

**Environment Setting** In our experiments, we follow the environment setting:

*Table 4.* Experiment Environment Details

| Component | Details |
| --- | --- |
| GPU | NVIDIA GeForce RTX 3070 |
| Python Version | 3.7.6 |
| Numpy Version | 1.21.6 |
| Pandas Version | 1.3.5 |
| Torch Version | 1.13.1 |
| Operating System | Linux |
| Processor | x86 64 |
| Architecture | 64-bit |
| Logical CPU Cores | 32 |
| Physical CPU Cores | 24 |
| Total Memory (GB) | 62.44 |

# I. Biological Interpretation of k-hop Convergence

A faster decay in the spectral norm implies that higher-order information becomes less influential, indicating a more localized brain interaction pattern, while a slower decay suggests that longer-range dependencies are preserved in the FC network.

As the k-value increases, more distant neural connections are considered, which may reflect the integration process of long-distance FC in the brain. Thus, the spectral norm convergence analysis provides quantitative insights into the scale and extent of connectivity alterations, complementing the interpretability of our tree-structured model.

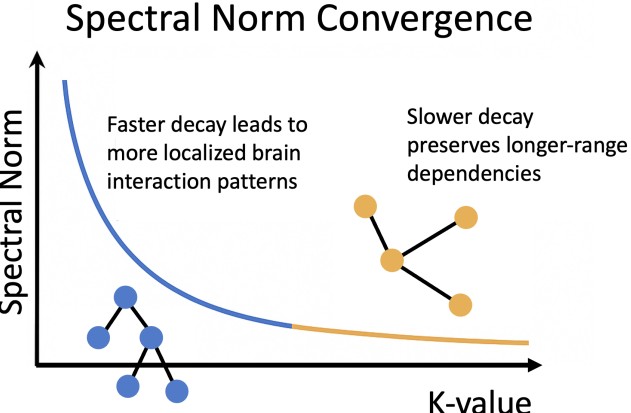

*Figure 9.* **The visualization of different decay meanings in spectral norm convergence.** The left side of the curve (blue section) demonstrates faster decay, which leads to more localized brain interaction patterns, visualized by a compact, hierarchical network structure with blue nodes. The right side (orange section) shows slower decay, which preserves longer-range dependencies, illustrated by a more distributed network with orange nodes connected across greater distances.

## J. Age-Related Brain Network Changes in Addiction and Schizophrenia

In this section, we identified age-related FC subnetwork changes associated with mental disorders, as shown in Fig. 10. his study reveals distinct patterns of brain network connectivity across different age groups in individuals with addiction and schizophrenia through FC analysis. In addiction patients, age emerges as a critical variable significantly influencing brain network organization. Young addiction patients (18-25 years) demonstrate strong connectivity between the DMN, VN, and FPN, which may reflect the collective influence of these critical networks during the early stages of addiction development. As age increases, middle-aged addiction patients (25-40 years) show connectivity extending to the DAN, indicating enhanced attentional control mechanisms during persistent addiction. Notably, older addiction patients (40-55 years) predominantly exhibit Somatomotor network connectivity, possibly representing specific effects of long-term addiction on motor control systems. In contrast, schizophrenia patients display different age-related network change trajectories. Young schizophrenia patients (18-25 years) are primarily characterized by strong connections between Somatomotor and VN, which may relate to abnormal sensory-motor integration in early schizophrenia. Middle-aged schizophrenia patients (25-40 years) demonstrate stronger FPN activity, suggesting the importance of higher-order cognitive control networks in disease progression. Older schizophrenia patients (40-66 years) are mainly characterized by enhanced connectivity between DAN and VN, potentially reflecting changes in attentional processing and visual integration mechanisms in late-stage disease. From our research findings, we demonstrate significant heterogeneity across different age groups in various mental disorders, particularly in the progression patterns of age-dependent network changes.

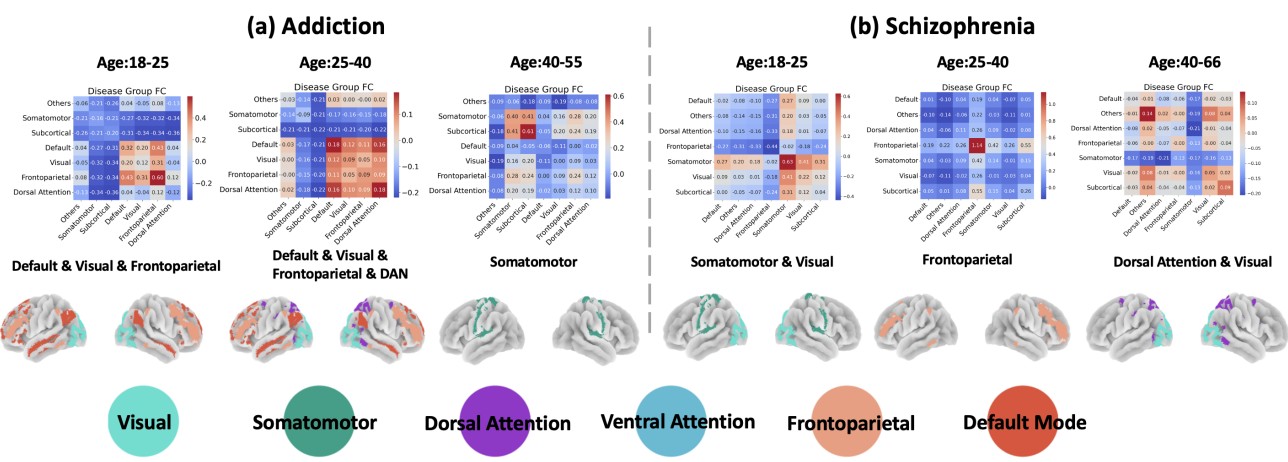

*Figure 10.* Predicted FC changes in different age groups and affiliated functional subnetworks.

## K. Effects of Age and CMFC Loss on Brain Tree Structure

According to Fig. 11, without parameter age and CMFC loss, the tree branches appear disorganized and fragmented, with paths lacking anatomical continuity and interpretability. However, with parameter that dynamically regulates FC based on individual age, it presents more coherent branches and clearer hierarchical structure.

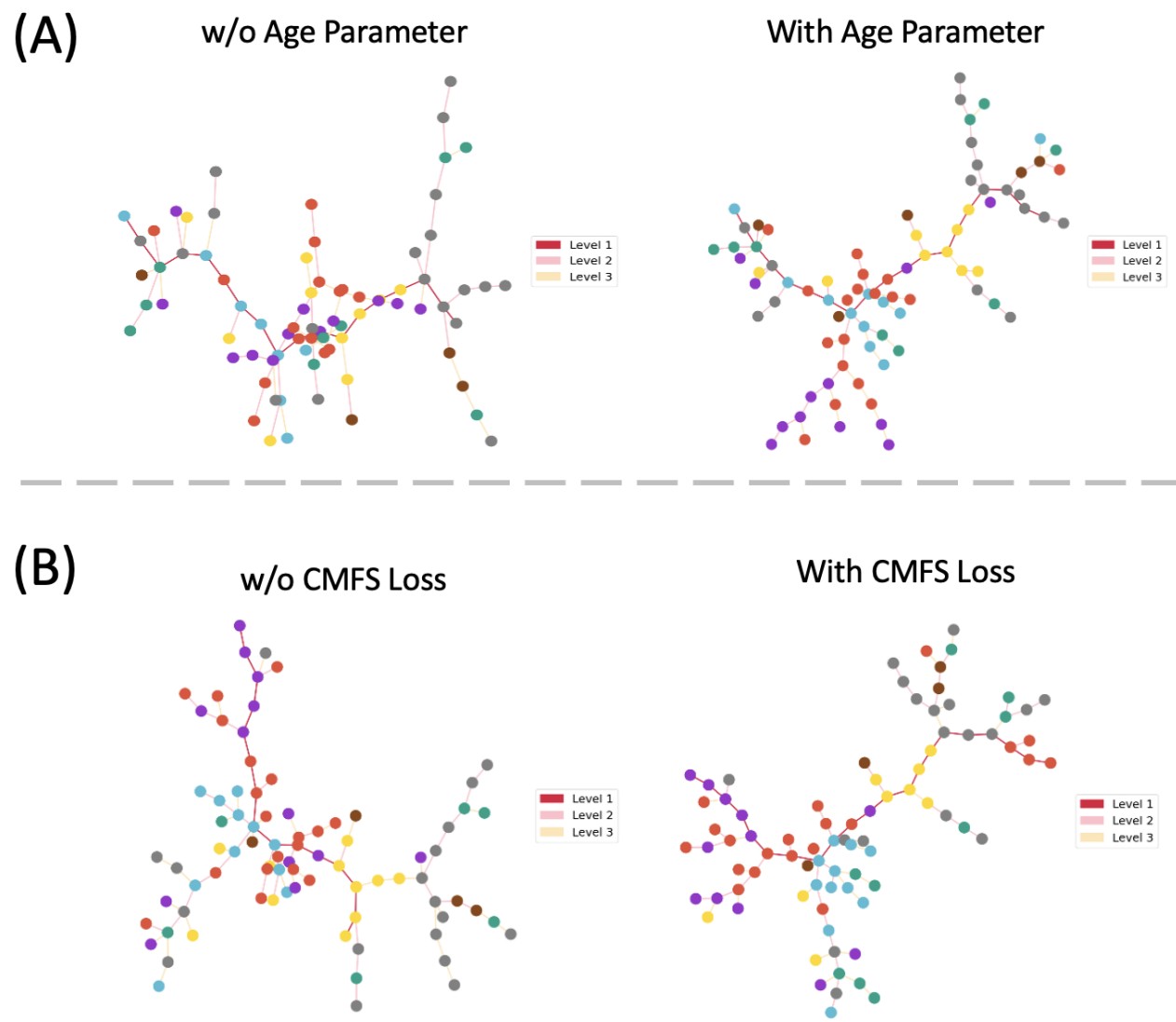

*Figure 11.* **Constructed brain Tree based on w/o θ and w/o CMFC loss.** We present different modeling approaches for brain connectivity networks and their robust in brain tree construct. (A) The left panel illustrates brain connectivity network models without age parameters, while the right panel demonstrates models incorporating age parameters. (B) The left panel displays brain connectivity network models without CMFS loss function, whereas the right panel shows models utilizing CMFC loss function.

## L. Dataset

We presents the summary statistic of two public fMRI dataset below:

*Table 5.* Summary statistics of demographics in Cannabis and COBRE

| Dataset | Sample size | Age (Mean ± SD) |
|---------|-------------|-----------------|
| Cannabis | HC (n=128) | 30.06 ± 10.47 |
| | Cannabis (n=195) | 27.23 ± 7.73 |
| COBRE | HC (n=72) | 38.31 ± 12.21 |
| | Schizophrenia (n=70) | 36.04 ± 13.02 |

### L.1. fMRI preprocessing in cannabis dataset

**fMRI Data Processing Pipeline** (Kulkarni et al., 2023) The fMRI data underwent a comprehensive preprocessing pipeline. Raw data was acquired using 3T MRI scanners (Siemens Trio or Philips) with T2*-weighted and high-resolution T1-weighted structural images. The preprocessing, implemented through FMRIPREP, included intensity non-uniformity correction, skull stripping, and brain surface reconstruction using FreeSurfer. Spatial normalization to the ICBM 152 template was performed, followed by brain tissue segmentation into cerebrospinal fluid (CSF), white matter, and gray matter. The functional data underwent slice-timing correction, motion correction, and co-registration to corresponding T1w images using boundary-based registration. Physiological noise was addressed through CompCor regression. Additional denoising steps included removing participants with framewise displacement > 3mm in over 5% of scan length and applying ArtRepair for remaining high-motion volumes. Combined task/nuisance regression was performed using motion parameters and aCompCor as covariates. The processed whole-brain time series were then parcellated using the Stanford functional ROIs atlas, resulting in 90 regions of interest. Quality control analyses confirmed that motion metrics did not significantly differ between groups (t-test p=0.86) and showed no correlation with classification accuracy (p=0.475).

## L.2. Different Atlases According to Seven Affiliated Functional Subnetworks

*Table 6.* Comparison of Brain Networks between Stanford (90 ROIs) and Harvard-Oxford (118 ROIs) Atlases

| Network | Stanford (90 ROIs) | Harvard-Oxford (118 ROIs) |
|---|---|---|
| Default Mode | bilateral mPFC, L/R lateral angular gyrus, R superior frontal gyrus, bilateral posterior/middle/mid-posterior cingulate, bilateral anterior thalamus, L/R parahippocampal gyrus, L/R inferior/mid-temporal cortex, L medial angular gyrus, L crus cerebellum (lang), L inferior parietal cortex, L/R middle frontal gyrus (vent_DMN), R inferior cerebellum (vent_DMN) | Left/Right Frontal Pole, Superior Frontal Gyrus, Middle Temporal Gyrus (anterior/posterior), Angular Gyrus, Precuneous Cortex |
| Visual | L/R mid occipital cortex, L/R precuneus (post_sal), bilateral medial posterior/medial precuneus, R angular gyrus - precuneus, L/R ventral precuneus, L/R fusiform gyrus, L/R mid-occipital cortex | Left/Right Middle/Inferior Temporal Gyrus (temporooccipital), Lateral Occipital Cortex (superior/inferior), Cuneal Cortex, Lingual Gyrus, Temporal Occipital Fusiform Cortex, Occipital Fusiform Gyrus, Occipital Pole |
| Dorsal Attention | L/R superior frontal gyrus, L/R superior/inferior parietal cortex, L/R precentral/fronto-opercular region, L/R inferior temporal cortex, L/R cerebellum (visuospatial), R crus cerebellum | Left/Right Middle Frontal Gyrus, Superior Parietal Lobule, Supramarginal Gyrus (anterior/posterior) |
| Frontoparietal | L/R middle frontal/dlPFC, L inferior frontal (triangular), L/R inferior parietal/angular gyrus, L inferior temporal gyrus, R/L crus cerebellum (LECN/RECN), L posterior thalamus, R middle orbito-frontal cortex, R superior medial frontal gyrus, R caudate | Left/Right Inferior Frontal Gyrus (pars triangularis/opercularis), Frontal Operculum Cortex |
| Somatomotor | L/R superior temporal/auditory, R thalamus, L/R pre/post-central gyri, bilateral supplementary motor area, L/R ventral posterior nucleus of thalamus, cerebellar vermis | Left/Right Precentral Gyrus, Superior Temporal Gyrus (anterior/posterior), Postcentral Gyrus, Juxtapositional Lobule Cortex, Central Opercular Cortex |
| Subcortical | R striatum/thalamus (2), L/R inferior frontal gyrus, pons/dropout region | Left/Right Thalamus, Caudate, Putamen, Pallidum, Brain-Stem, Hippocampus, Amygdala, Accumbens |
| Others | L prefrontal cortex, L/R anterior insula, bilateral ACC, L/R crus cerebellum (ant_sal), L/R middle frontal gyrus, L/R supramarginal cortex, L/R middle thalamus, L/R anterior cerebellum, L/R superior temporal gyrus, bilateral calcarine cortex, L LGN | Left/Right Insular Cortex, Temporal Pole, Frontal Medial/Orbital Cortex, Subcallosal Cortex, Paracingulate Gyrus, Cingulate Gyrus (anterior/posterior), Parahippocampal Gyrus, Temporal Fusiform Cortex, Parietal Operculum Cortex, Planum regions, Cerebral White Matter/Cortex, Lateral Ventricle |

