# OpenReview forum: "NeuroTree: Hierarchical Functional Brain Pathway Decoding for Mental Health Disorders"
_ICML.cc/2025/Conference — ICML 2025 poster_

### Official Review · Reviewer_gVcG · 2025-03-13

**Overall Recommendation:** 4

**Summary:**

The paper introduces Neurotree, a graph convolutional network that employs ordinary differential equations (ODEs) to model neural dynamics and learns a tree topology using contrastive loss to identify functional connectivity (FC) pathways. The model is evaluated on two datasets, achieving state-of-the-art performance. The primary contribution lies in its ability to characterize differences in functional connectivity between patients and healthy subjects, offering insights into neural circuit disruptions in psychiatric disorders.

**Claims And Evidence:**

- The claim that "addiction may lead to stronger brain connectivity signals compared to patients with psychiatric disorders" lacks justification. The authors should clarify why this is intuitive and provide supporting evidence. Perhaps I missed this.

- The authors claim to model causal graph structures but construct a weighted undirected graph as mentioned "we construct a weighted undirected graph". Why was directed graph not used and how can we identify causal relationships instead? Is this based on the tree hierarchy?

**Essential References Not Discussed:**

No essential references appear to be missing. However, the authors could consider expanding on advancements in graph-based neural network architectures for brain connectivity analysis.

**Experimental Designs Or Analyses:**

- An ablation study was conducted on contrastive masked FC strengths and age modulation as inputs. However, the necessity of analyzing the rate of spectral norm decrease with k-hop is unclear. The authors should further elaborate on the insights gained from this analysis. What does spectral norm mean and why are we interested in K-hop convergence?

- Table 1 shows a significant performance improvement when age is included as an input. The authors should explain why age contributes to performance and clarify whether previous methods also used age as an input. Additionally, if age is provided during training, how was it predicted for age groups in Table 2?

- The contrastive masked FC strength (CMFS) loss does not significantly improve performance in Table 1. Is this loss more critical for learning the functional hierarchy in Figure 4? How does the brain tree structure change without θ and CMFS loss? Perhaps additional figures similar to Fig. 4 could be included in the appendix with different objectives.

- Could directed graphs be constructed using the temporal $A^d(t)$ matrix? If so, how would this affect the formation of the brain tree network?

**Methods And Evaluation Criteria:**

- The model was evaluated on two datasets, but the training process was not thoroughly described, leaving gaps in reproducibility. The authors said that they will release the code upon acceptance which should address this.

- Neurotree was compared against two baseline models and four state-of-the-art models using classification accuracy as the primary metric. While the results are promising, the choice of evaluation metrics could be expanded to include additional performance measures, such as the loss training and test curves, including when adding age as an input $\theta$ and CMFS objective. Alternatively, more details on the dataset and how the corresponding performance metrics e.g. AUC, Acc, Prec, Rec makes sense in this context should be discussed.

- The authors used Yeo's 7 parcellations to construct the brain tree. It is unclear whether other parcellation schemes were considered and whether they yield similar hierarchical structures.

**Other Comments Or Suggestions:**

No additional comments or suggestions.

**Other Strengths And Weaknesses:**

- Originality: While the framework builds on existing ideas, the theoretical improvements and the ability to visualize brain tree structures are notable contributions.
- Significance: The model's ability to elucidate functional hierarchies in brain networks is appealing and could have broad applications in neuroscience.
- Clarity: The paper is generally well-written, but some sections, such as the training process and the rationale behind certain analyses, could be clarified.

**Questions For Authors:**

- How does the brain tree structure change when θ and CMFS loss are omitted during network construction?
- Have the authors considered using alternative parcellation schemes to verify the robustness of the brain tree hierarchy predictions?
- Could the authors elaborate on the insights gained from analyzing the rate of spectral norm decrease with k-hop?
- How does the inclusion of age as an input during training affect the model's ability to predict age groups in Table 2? Was age also provided as an input during prediction?

**Relation To Broader Scientific Literature:**

Neurotree is highly relevant to computational psychiatry and neuroscience. Its application to other datasets and signal modalities, such as EEG or multi-region electrophysiology, could further validate its utility. Additionally, the framework could be adapted to study other network types, such as social or power networks.

**Theoretical Claims:**

I reviewed the theoretical considerations but did not perform a thorough verification. The proposed bounds and theorems appear valid, though a more detailed examination would be necessary to confirm their correctness.

---

> ### Author Rebuttal · Authors · 2025-03-30
>
> > **C1. Cannabis addicts have stronger FC compared to schizophrenia patients.**
>
> We appreciate the reviewer's careful inquiry about this claim! Study [1] found cannabis users show higher baseline functional connectivity in reward circuits than schizophrenia patients. Our `Fig 4. (a-2) and (a-3)` results predict higher FC numbers in cannabis users across Yeo's seven brain networks, with additional FC degree centrality predictions available at the link:https://anonymous.4open.science/r/anonymous_ICML/anonymous_DC.png.
>
> References:
> [1]Fischer et al. Journal of Schizophrenia Research, 2014.
>
> > **M3 & Q2.: Other parcellation schemes methods and robust approaches using NeuroTree.**
>
> Thank you for your valuable comment!
>
> `1). Other parcellation schemes`: The addiction and COBRE atlases contain 90 and 118 ROIs respectively. As shown in `section 7 and Fig 4`, regardless of which atlas was used, the extracted tree-structured pathways successfully mapped to Yeo's 7 networks, enabling cross-dataset interpretation. Our framework consistently revealed distinct hierarchical structures in psychiatric conditions across both parcellations, demonstrating the stability and generalizability of our tree construction method. In future work, NeuroTree can easily extend to additional parcellations (e.g., AAL, BASC) for different disease research problems.
>
> `1). The robust of NeuroTree`:  As described in `Section 6.1` and shown in Table 1 with SOTA performance, despite differences in parcellation schemes, we can improve stability through training the NeuroTree framework to obtain nodes (regions) prediction and weighted paths. This indicates that the hierarchical brain tree structures constructed by NEUROTREE are robust to the choice of ROI definitions. According to Definition 3.5, Kruskal's algorithm ensures tree decomposition with the shortest paths.
>
> > **E1 & E3 & M2: NeuroTree uses the CMFS loss model performance and the biological significance of k-hop using spectral norm**
>
> We appreciate the reviewer’s insightful question! `1). Loss ablation`:  According to the experiment w/o CMFS loss in `Table 1`, we found that adding CMFS loss can make the overall (+ CE loss) loss more robust in our tensorboard plot at link:https://anonymous.4open.science/r/anonymous_ICML/anonymous_loss_comparison.png.
>
> `2.) K-hop convergence`: The intuition for analyzing the rate of spectral norm decay with increasing k-hop allows us to examine how rapidly information from distant nodes attenuates across the dynamic brain network structure. A faster decay in spectral norm indicates more localized brain interactions as higher-order information becomes less influential. Conversely, slower decay preserves longer-range dependencies in the FC network. Higher k-values incorporate more distant neural connections, potentially reflecting the brain's long-distance functional integration processes.
>
> > **M2 & Q1 & Q4:How does the inclusion of age as an input during training affect the model's ability to predict age groups in Table 2? Was age also provided as an input during prediction?**
>
> We appreciate your thoughtful review.
> `1.) Effectiveness of including age variable in the model`: Recent research [2,3] shows incorporating demographic data (especially age) into GNN for fMRI enables more precise learning of subject differences. NeuroTree's ODE design uses age to regulate features during message passing, enhancing graph classification through age-aware GCN. This approach accounts for individual fMRI differences, demonstrating how personal features like age can stabilize dynamic graph convolutional neural network performance.
>
> References:
>
> [2]Zhang, Hao, et al. IEEE TMI, 2022
>
> [3]Wang, Xuesong, et al. MICCAI, 2022.
>
> `2.) Effects of age and CMFS loss on tree structure`:
> According to our supplementary figure at link https://anonymous.4open.science/r/anonymous_ICML/anonymous.png, without parameter $\theta$ and CMFC loss, the tree branches appear disorganized and fragmented, with paths lacking anatomical continuity and interpretability. However, with parameter $\theta$ that dynamically regulates functional connectivity based on individual age, it presents more coherent branches and clearer hierarchical structure.
>
> `3.) Is age included as a prediction in Table 2?`:
> The advantage of NeuroTree is that it is age-aware GCN to learn the influence of age on dynamic FC patterns. We follow the currently existing literature [2,3] conventional practices, incorporating age information as part of the model input for feature learning. We input age during the training phase, `but to avoid data leakage, we do not include age in the prediction phase.`
>
> We supplement that the complete results of Table 2 the `NeuroTree can also be used in the training and testing process without including age` at the link:https://anonymous.4open.science/r/anonymous_ICML/anonymous_table.png.
>
> **Thank you for your thoughtful review. We've addressed each question as fully as possible within the word limit.**

---

### Official Review · Reviewer_XciT · 2025-03-16

**Overall Recommendation:** 1

**Summary:**

This paper proposes NeuroTree as a framework for feature learning from functional connectivity for brain disease characterization. NeuroTree integrates standard graph convolutional network with neural ordinary differential equations.

**Claims And Evidence:**

I find various claims and evidence in this paper problematic.

1. The paper claims to integrate `interpretable variables into both static and dynamic causal graph convolutional modeling'. Here, it's very unclear how the `interpretability' of a specific variable is gauged. Moreover, I do not see any convincing statistical evidence of causal modeling or its evaluation within this paper.

2. The proposed model achieves the maximum mean classification accuracy of 73% at best among all experiments considered. Therefore, it is clearly not an effective predictor of brain disease. I understand that the authors claim improvements over other methods. However, if 73% is the best you can achieve on given datasets as your main result, then there is perhaps a clear mismatch between the considered model and the task at hand.

3. I also find the results within the section 'Chronological Brain Age Prediction' problematic. Firstly, chronological age and brain age can differ in populations with brain diseases (which forms the primary motivation for this line of research). Therefore, the authors should rigorously clarify their interpretation and definition of 'Chronological Brain Age'. Furthermore, the gap between chronological age and brain age is often the biomarker, and predicting chronological age within disease cohorts with high Pearson correlation has no practical utility.

**Essential References Not Discussed:**

The paper completely ignores the literature on graph convolutional networks from the lens of graph signal processing, and their applications as age prediction models, fMRI characterization, and interpretable biomarker construction. I am not naming specific studies here but I would recommend that the authors review this line of literature as well.

**Experimental Designs Or Analyses:**

See Claims And Evidence section.

**Methods And Evaluation Criteria:**

See Claims And Evidence section.

**Other Comments Or Suggestions:**

I don't have any other comments.

**Other Strengths And Weaknesses:**

See Claims And Evidence section for weaknesses.

**Questions For Authors:**

N/A

**Relation To Broader Scientific Literature:**

The paper lacks solid conceptual contributions relative to broader scientific literature.

**Theoretical Claims:**

I did not check the correctness of theoretical claims.

---

> ### Author Rebuttal · Authors · 2025-03-31
>
> **We thank the reviewer for valuable suggestions and insightful comments, and we have clarified a few things about accuray and brain age so that our contribution can be better understood.**
>
> > **Q1. About the terminology of 'interpretability' and 'causal' modeling.**
>
> `1.) The three meanings of interpretability in NeuroTree:`
>
> * In this work, we integrate the deep learning model concept from the literature [1,2,3] to enhance brain disorders classification. We define "interpretable variables" as observable latent factors in our model, not only providing an fMRI modeling but also showing the level of variations. We have corrected our terminology to "demographics" in the revised manuscript.
>
> * In section 3.1, we integrate age as parameter $\theta$ into ODE-GCN and  predict disease-relevant regions, visualized through node importance scores (brain regions) in tree explanations, enhancing model interpretability (Fig. 4).
>
> * NeuroTree can completely perform model training to hierarchically decompose the fMRI brain network into a tree structure, with each level showing significant brain regions on the tree path to help explain brain disease, which provides further interpretability.
>
> `2.) Statistical Interpretation:` We will conduct hypothesis testing through alation study, comparing our framework with ODEs and without ODEs to show the statistical significance of using causal network analysis.
>
> `3.) The definition of 'causal' modeling:`  We clarify that our use of the term "Causal" originates from explanations in deep learning literature [4,5]. For example, dynamic causal graph learning in [4] enables neural networks to automatically learn "time-varying causal graphs" or causal structures from traffic data. In Section 3.1, NeuroTree employs the ODE model form Eq (1) to simulate influences between brain regions, reflecting a continuous-time causal dynamic relationship. Although we have not yet used traditional Granger causality or dynamic causal modeling to directly present a directed graph node method, the time-varying graphs (dynamic FC) derived from ODE provide a framework that approximates causal relationships. We appreciate the reviewer's perspective and we have revised the manuscript to remove the "causal" word to make our paper more clear.
>
> [1]Zheng, et al. "Brainib: `Interpretable` brain network-based psychiatric diagnosis with graph information bottleneck." IEEE TNNLS, 2024.
>
> [2]Cui, et al. "`Interpretable` graph neural networks for connectome-based brain disorder analysis." MICCAI, 2022.
>
> [3]Chen, et al. "Learnable subdivision graph neural network for functional brain network analysis and `interpretable` cognitive disorder diagnosis." MICCAI, 2023.
>
> [4]Lin, et al. "Dynamic causal graph convolutional network for traffic prediction." IEEE CASE. 2023.
>
> [5]Wein, et al. "A graph neural network framework for causal inference in brain networks." Scientific reports, 2021.
>
> > **Q2. About model accuracy in brain network classification and comparison of models.**
>
> We clarify this from two perspectives:
>
> Our model NeuroTree combines achieved the state-of-the-art performance using dynamic fMRI compared to similar models like PathNNs, BrainGNN, etc., given the variance in the individual fMRI data [8,9]. However, we believe this is an important yet essential step to predict the patients with mental disorders, especially since we also achieved an **AUC of 0.71** on the **public COBRE dataset**. In the future, NeuroTree can integrate with other modalities (e.g., DTI, genetic information), allowing us to examine the connection alterations among patients.
>
> [8]Zhao,et al. "Enhancing major depressive disorder diagnosis with dynamic-static fusion graph neural networks." IEEE JBHI, 2024.
>
> [9]Peng, et al. "Gate: Graph CCA for temporal self-supervised learning for label-efficient fmri analysis." IEEE TMI, 2022.
>
>
> > **Q3. About 'Chronological Brain Age Prediction' terminology definition problems and model prediction usefulness.**
>
> `1. Clarification of definition and interpretation:` In our work, chronological age refers to the subject’s actual age, which is used as a regulatory parameter to model age-related changes in FC. Specifically, in our AGE-GCN, we incorporate the age parameter $\theta$ to learn how FC strength evolves over time, thus enhancing the interpretability of aging patterns in mental disorders (See Eq. (3)–(12)).
>
> `2. The utility of predicting actual age in disease populations: ` We fully agree with the reviewer that the difference between chronological age and brain age is a clinically meaningful biomarker. However, NeoroTree is not solely designed to predict age accurately. Instead, it uses age as a modulation variable to explore how FC patterns vary with age across different clinical groups (e.g., cannabis users, schizophrenia patients).
>
> 3. In the revised version, we conduct the prediction of brain ages among healthy controls, while comparing the prediction accuracy from patients with addictive disorders.

---

### Official Review · Reviewer_fciF · 2025-03-23

**Overall Recommendation:** 3

**Summary:**

This paper introduces NEUROTREE, a novel framework for analyzing functional brain networks derived from fMRI data.  The framework integrates k-hop Graph Convolutional Networks (GCNs) with neural Ordinary Differential Equations (ODEs) to enhance the learning of dynamic functional connectivity (FC) features and capture high-order brain regional pathway features in a tree topology.  The authors demonstrate the effectiveness of NEUROTREE in predicting psychiatric disorders and elucidating their underlying neural mechanisms across two distinct mental disorder datasets.

**Claims And Evidence:**

Looks good to me.

**Essential References Not Discussed:**

N/A

**Experimental Designs Or Analyses:**

The authors may consider providing significance analysis as they claimed that their method significantly outperforms SOTA models.

**Methods And Evaluation Criteria:**

The authors only evaluated on two datasets, Cannabis and COBRE. While the use of publicly available datasets is commendable for reproducibility and comparison purposes, the selection is limited in scope with the following concerns: 1) limited disorder representation; 2) dataset heterogeneity; 3) lack of demographic diversity.

Additionally, the authors are encouraged to explicitly mention the sample sizes of the datasets used. Larger datasets would provide more robust validation and enhance the credibility of the findings.

**Other Comments Or Suggestions:**

Some writing issues:
1. Ln 043 - 045 (right): "Nevertheless, ... However..." The sentence begins with "Nevertheless" and then uses "However" shortly after. Both words serve a similar purpose (signaling contrast or limitation), so using both is repetitive and disrupts the flow.
2. Consider explaining what is sigma in Equ. 1.
3. Consider (1) clearly distinguishing between matrix and scalar values in equations, and (2) specifying the dimensions of each matrix.

**Other Strengths And Weaknesses:**

Strengths:
1. Novelty and technical soundness: NEUROTREE offers a novel approach by integrating k-hop GCNs with neural ODEs. It is well-explained, with clear descriptions of the k-hop ODE-GCN, Contrastive Masked FC Strength (CMFS) optimization, and hierarchical brain tree construction. The authors also provide theoretical support for their methods, including theorems and proofs in the appendices.
2. Interpretability: A key strength of NEUROTREE is its interpretability.  The framework facilitates the identification of hierarchical neural behavioral patterns and provides insights into age-related deterioration patterns, enhancing the understanding of underlying neural mechanisms in psychiatric disorders.

Weaknesses:
1. Complexity and computational cost: The proposed NEUROTREE framework is complex, involving multiple components and parameters.  This complexity may make it challenging for researchers to implement and apply the model. The use of k-hop GCNs and neural ODEs can be computationally expensive, potentially limiting the scalability of the model to larger datasets.
2. Limited generalization: The study focuses on two specific psychiatric disorders.
3. Low reproducibility: No code/models are provided.

**Questions For Authors:**

N/A

**Relation To Broader Scientific Literature:**

N/A

**Theoretical Claims:**

The authors make several theoretical claims, which appear to be correct. Theorem 3.2 posits that the l2-norm of the k-hop connectivity adjacency operator is bounded as k-hp approaches infinity. Theorem 3.4 describes the discretization of Age-Aware Continuous-Time Graph Convolution, building upon previous work by Tang et al. (2024). Detailed proofs are provided in the appendix.

---

> ### Author Rebuttal · Authors · 2025-03-29
>
> > **M1 \& W2:** **The study's evaluation is limited by using only two datasets, including disorder representation, dataset heterogeneity, and demographic diversity.**
>
> We thank the reviewer concern for data diversity!
> Due to data privacy concerns and the limited availability of public fMRI data for individuals with mental disorders that also include demographic information. **We believe our proposed NeuroTree method's contributions are beneficial not only for decoding specific disease fMRI datasets but also extend to broader research applications such as EEG in neuroscience.**
>
> > **M2:**  **Summary statistics of demographics in two datasets and sample size.**
> We thank the reviewer for taking the time to review our work. Due to page limitations, we have placed the demographic statistics details for both datasets in `Appendix H. Dataset`. The total Cannabis sample size is `323` and COBRE is `142`.
>
> > **W1:** **Model computational cost and complexity, and the scalability for future studies.**
>
> We appreciate for pointing out this issue.
> `1).Computational cost`:
> We have additionally supplemented experiments conducted for 100 epochs including a comparative table of computational costs with and without parameters ($\theta$,$\Gamma$,$\lambda$) across two datasets. To facilitate comparison of computational costs, we have included a Graph Transformer-based module with higher computational cost for comparison in the table below:
>
> | Dataset (Model)             | Type    | Training Time (sec) | GPU Memory (MB) | Inference Time (sec，avg over 10 runs) |
> |-----------------------------|--------------|----------------|------------------------|---------------------------|
> | Cannabis (NeuroTree)        | With params       | 13.958         | 20.62                  | 0.000870                  |
> |                             | NO params     | 5.396          | 14.25                  | 0.000160                  |
> | Cannabis (Graph Transformer)| With params       | 19.065         | 170.38                 | 0.002445                  |
> |                             | NO params     | 5.232          | 18.53                  | 0.000346                  |
> | COBRE (NeuroTree)           | With params       | 6.157          | 18.12                  | 0.001771                  |
> |                             | NO params     | 2.137          | 9.21                   | 0.000351                  |
> | COBRE (Graph Transformer)   | With params       | 8.948          | 213.08                 |0.002608                    |
> |                             | NO params     | 2.173          | 13.46                  | 0.000356                    |
>
> `2). Code Implementation`:
> In addition, the model we designed can be easily trained on a personal computer (include GPU, Google Colab Jupyter Notebook), the related environment setting can be found in our paper `Appendix G Table 4`. Compared to Graph Transformer modules with their extensive matrix calculations, NeuroTree significantly  reduces computational demands through strategic parameter configurations. Additionally, NeuroTree enables researchers to observe parameter effects, enhancing model interpretability in alignment with specific research objectives.
>
> > **W3:**  **Low reproducibility: No code/models are provided.**
> Thank you very much for the reviewer's discussion on the reproducibility of our research!
>
> `1). Code acquisition`: We will release the complete code (including tree plot) and processed fMRI data on GitHub at the camera-ready stage when the paper is accepted.
>
> `2). Reproducible`:
> We demonstrate relevant links to the training and testing process in Tensorboard according to the reviewer's (**gVcG**) suggestion for your reference:https://anonymous.4open.science/r/anonymous_ICML/anonymous_loss_comparison.png
>
> > **O1:** **Redundant Sentence Correction.**
> We appreciate your careful review and corrections! We express our sincere apologies regarding the extra sentence mistake. We have corrected the English grammar expression to maintain logic.  *However, current approaches face two fundamental limitations. First, .. Second,..*
>
> > **O2 \&  O3:** **Consider (1) clearly distinguishing between matrix and scalar values in equations, and (2) specifying the dimensions of each matrix.**
> We thank the reviewer for the detailed review of our work!
> We have added explanations for $\sigma$ being the sigmoid function in Eq (1) and $\eta, \rho \in  \mathbb{R}^+$ for the scalar values in Eq (2) of the paper. We clarify that the external input vector $u(t) \in \mathbb{R}^{v}$ interacts with the external stimulus encoding matrix $C \in \mathbb{R}^{v \times v}$, while both the static adjacency matrix $A^{s} \in \mathbb{R}^{v \times v}$ and the time-dependent dynamic adjacency matrix $A^{d}(t) \in \mathbb{R}^{v \times v}$ contribute to the overall network connectivity at time $t$, and $D^{-\frac{1}{2}} A D^{-\frac{1}{2}} H^{(l-1)} W^{(l-1)} \in \mathbb{R}^{v \times d_{\text{out}}}$ with output $d$ dimension.

---

### Decision · Program_Chairs · 2025-05-01

**Decision:**

Accept (poster)

**Comment:**

This paper received one accept, one weak accept, and one reject. While one reviewer raises several concerns, including the lack of strong justification for focusing on psychiatric disorders, limited experimental validation, and unclear interpretations of "chronological brain age," the paper also introduces technically novel and interpretable methods. The proposed NEUROTREE framework, combining k-hop GCNs with neural ODEs, offers a fresh and theoretically grounded approach to modeling hierarchical brain structures. Its focus on interpretability and age-related neural patterns is particularly valuable. Although computational complexity and limited generalization are valid concerns, and reproducibility could be improved with code release, the Area Chair finds that the core issues are largely addressable. Given the interdisciplinary relevance of the work to neuroscience applications, the authors are encouraged to revise the final version to address the reviewers’ concerns and clarify the conceptual framing and empirical contributions.